

# Molecular-level evidence for marine aerosol nucleation of iodic acid and methanesulfonic acid

An Ning[1], Ling Liu[1], Lin Ji[2], and Xiuhui Zhang[1]

[1]Key Laboratory of Cluster Science, Ministry of Education of China, School of Chemistry and Chemical Engineering, Beijing Institute of Technology, Beijing 100081, China

[2]Department of Chemistry, Capital Normal University, Beijing 100048, China

*Correspondence to*: Xiuhui Zhang (zhangxiuhui@bit.edu.cn)

**Abstract.** Both iodic acid ($HIO_3$, IA) and methanesulfonic acid ($CH_3S(O)_2OH$, MSA) have been identified by field studies as important precursors of new particle formation (NPF) in marine areas. However, the mechanism of NPF in which IA and MSA are jointly involved is still unclear. Hence, we investigated the IA-MSA nucleation system under different atmospheric conditions and uncovered the corresponding nucleating mechanism at the molecular level for the first time, using quantum chemical approach and Atmospheric Cluster Dynamics Code (ACDC). The findings showed that MSA can stabilize IA clusters via both hydrogen and halogen bonds. Moreover, the joint nucleation rate of IA-MSA system is significantly higher than that of IA self-nucleation, particularly in relatively cold marine regions with sparse IA and rich MSA. For the IA-MSA nucleation mechanism, in addition to self-nucleation of IA, the IA-MSA-involved clusters can also directly participate in the nucleation process, and their contribution is particularly prominent in the polar regions with rich MSA and sparse IA. The IA-MSA nucleation mechanism revealed in this work may help to elucidate some missing sources of marine NPF.

## 1 Introduction

Marine aerosols, being the primary natural aerosol system (O'dowd and De Leeuw, 2007), significantly affect global radiation balance and climate by regulating cloud properties as cloud condensation nuclei (CCN) (Takegawa et al., 2020; IPCC, 2013). Nearly half of the CCNs originate from the new particle formation (NPF) via the gas-to-particle conversion (Merikanto et al., 2009; Yu and Luo, 2009). As a major source of CCN globally, NPF mainly consists of the nucleation of gaseous molecules and the subsequent growth of the formed clusters (Kulmala et al., 2013; Kulmala, 2003; Zhang, 2010). Although extensive studies have provided observational evidence of frequent NPF events in the coastal zone, open ocean, and even ice-covered polar regions (Zheng et al., 2021; Takegawa et al., 2020; Sipila et al., 2016; O'dowd et al., 2002; Baccarini et al., 2020), the corresponding NPF mechanisms at the molecular level remain poorly understood stemming from the lack of chemical speciation in the initial nucleating steps.

Marine NPF is more affected by biological emissions compared to inland ones with anthropogenic influence, particularly in remote marine areas (Kerminen et al., 2018). Historically, sulfur-containing species originating from ocean-emitted dimethyl





sulfide (DMS) have long been identified as significant components of marine aerosols (Charlson et al., 1987; Shaw, 1983; Bates et al., 1992). Methanesulfonic acid ($CH_3S(O)_2OH$, MSA), as a well-known oxidation product of DMS (Chen et al., 2018; Hatakeyama et al., 1982), is widely dispersed throughout the world's oceans and has considerable atmospheric concentrations (Chen et al., 2018), comparable to or higher than sulfuric acid (SA), [MSA]/[SA] = 10%−250% (Berresheim et al., 2002; Davis et al., 1998; Eisele and Tanner, 1993). Moreover, MSA has been experimentally demonstrated to be a significant nucleating

precursor in coastal and remote oceans (Dawson et al., 2012; Hodshire et al., 2019; Karl et al., 2007). Along with stricter global controls on anthropogenic $SO_2$ emissions, the impact of MSA on NPF will become increasingly significant in the future (Perraud et al., 2015), particularly in marine areas.

In addition to the above sulfur precursors, recent experimental and theoretical studies (He et al., 2021; Martín et al., 2020; Xia et al., 2020; Rong et al., 2020) have also recognized the critical role of iodine compounds in marine NPF process.

According to field research (Sipila et al., 2016), the observed intense NPF events occur during low tide in the coastal Mace Head and are accompanied by a significant increase in iodic acid ($HIO_3$, IA) concentration, indicating that the coastal NPF is primarily driven by IA self-nucleation. Also, such a high association between IA and NPF events also exists in other marine regions (Baccarini et al., 2020; Beck et al., 2020). More recently, evidence suggests that the NPF events in the sea-ice covered Arctic region are mainly driven by IA (Baccarini et al., 2020). Notably, in addition to IA, significant concentrations of MSA

were observed during marine NPF events (Beck et al., 2020). Although MSA and IA are both found in the particle phase (Beck et al., 2020), it is still unknown whether they could be simultaneously involved in the early nucleation process. If so, their joint nucleation mechanism and the corresponding regions affected by that mechanisms need be further elucidated.

Herein, high-level quantum chemical calculations combined with Atmospheric Clusters Dynamic Code (ACDC) (Mcgrath et al., 2012) were employed to simulate the nucleating process of the $(IA)_x \cdot (MSA)_y$, ($0 \leq x \leq 6$, $0 \leq y \leq 3$, $1 < x + y \leq 6$)

system. Under different atmospheric conditions (temperature and precursor concentration), a series of ACDC simulations were carried out to explore: i) the binding nature of IA and MSA, ii) the joint effects of IA and MSA on the nucleation, iii) which locations are more affected by the IA-MSA mechanism. The current work may contribute to developing a more comprehensive marine NPF mechanism and explaining some missing sources of particles at marine regions.

## 2 Methods

### 2.1 Quantum chemistry calculations

All structure optimizations with tight convergence criteria and frequency calculations were carried out with Density function theory (DFT) using Gaussian 09 package (Frisch et al., 2009). Considering the variety of possible isomers of multimolecular clusters, a systematic multi-step conformation search (Ning et al., 2020) was employed here to locate the lowest-energy cluster structures. For each studied IA-MSA cluster, the artificial bee algorithm combining the CHARMM (Mackerell Jr et al., 1998)

force field was adopted to yield 1000 initial configurations by ABCluster software (Zhang and Dolg, 2015). After pre-





optimization by the PM7 semi-empirical method (Stewart, 2013) with MOPAC2016 (Stewart, 2016), the structures with low energy were left for further optimization. The final global minima were reoptimized using $\omega$B97X-D functional due to its best performance in studying atmospheric clusters (Elm and Kristensen, 2017; Schmitz and Elm, 2020), where the 6-311++G(3df,3pd) (Francl et al., 1982) basis set was selected for H, C, O and S atoms, and aug-cc-pVTZ-PP with ECP28MDF

for I atom (Peterson et al., 2003).

The single-point correction was further performed by RI-CC2 method (Hattig and Weigend, 2000) with aug-cc-pVTZ (for H, C, O) + aug-cc-pV(T+d)Z (for S) + aug-cc-pVTZ-PP with ECP28MDF (for I) basis set using TURBOMOLE program (Dunning et al., 2001; Ahlrichs et al., 1989), since the ACDC simulations based on RI-CC2 values are in good agreement with the experimental results (Lu et al., 2020; Kürten et al., 2018; Li et al., 2020; Almeida et al., 2013). Herein, in this work, the

Gibbs formation free energies ($\Delta G$, kcal mol$^{-1}$) of the studied clusters were calculated as Eq. (1):

$$\Delta G = \Delta E_{\text{RI-CC2}} + \Delta G_{\text{thermal}}^{\omega\text{B97X-D}} \tag{1}$$

where $\Delta E_{\text{RI-CC2}}$ is the electronic contribution and $\Delta G_{\text{thermal}}^{\omega\text{B97X-D}}$ is the thermal contribution to Gibbs free energy. For subsequent clustering kinetic simulations at different temperatures, the $\Delta G$ of clusters ranging from 218 K to 298 K were calculated by Shermo 2.0 (Lu and Chen, 2021) and collected in Table S2.

**2.2 Wavefunction analysis**

To better understand the interactions between IA and MSA, the bonding nature was investigated through wave function analysis using Multiwfn 3.7 (Lu and Chen, 2012). Specifically, the electrostatic potential (ESP) on the molecular was calculated for IA and MSA, which facilitates understanding their potential interaction sites. Moreover, the natural bond orbital (NBO) analysis (Reed et al., 1988) was carried out to give a detailed insight into intermolecular interactions. Based on the

final identified clusters, the NBO information calculated by Gaussian 09 is resolved by Multiwfn and the key interactive orbitals are visualized by VMD 1.9.3 (Humphrey et al., 1996). To further quantify the binding strength, electron density $\rho(r)$, Laplacian electron density $\nabla^2\rho(r)$, energy density $H(r)$ at corresponding bond critical points (BCPs) based on atoms in molecules (AIM) theory (Becke, 2007; Lane et al., 2013) were also calculated in this work (Table S1 in ESI†).

**2.3 Atmospheric clusters dynamic simulations**

Simulation of the nucleation process of the IA-MSA system is achieved by the Atmospheric Clusters Dynamic Code (ACDC) (Mcgrath et al., 2012). Specifically, the ACDC derives the steady-state concentration and cluster formation rates by solving the birth-death equations (Eq. (2)).

$$J = \frac{dc_i}{dt} = \frac{1}{2}\sum_{j<i}\beta_{j,(i-j)}C_jC_{(i-j)} + \sum_j\gamma_{(i+j)\rightarrow i}C_{i+j} - \sum_j\beta_{i,j}C_iC_j - \frac{1}{2}\sum_{j<i}\gamma_{i\rightarrow j}C_i + Q_i - S_i \tag{2}$$





where $C_i$ refers to the cluster $i$ concentration, $\beta_{i,j}$ is the collision coefficient between clusters $i$ and $j$, $\gamma_{i \to j}$ is the evaporation
coefficient of smaller cluster $j$ from the parent cluster $i$, $Q_i$ and $S_i$ are the outside source and loss term of cluster $i$, ,
respectively. $\beta_{i,j}$ is calculated based on the kinetic gas theory (Seinfeld and Pandis, 2006), which is given as:

$$\beta_{i,j} = \left(\frac{3}{4\pi}\right)^{1/6} \left(\frac{6k_B T}{m_i} + \frac{6k_B T}{m_j}\right)^{1/2} \left(V_i^{1/3} + V_j^{1/3}\right)^2 \qquad (3)$$

where $V_i$ and $m_i$ are the volume and mass of cluster $i$, respectively. $k_B$ is t Boltzmann constant, and $T$ is the temperature. Eq.(3)
is derived from the hard-sphere collision theory where $V_i = 3/4 \times \pi \times (D_i/2)^3$. The diameter $D_i$ of cluster $i$ is calculated by
Multiwfn (Lu and Chen, 2012). Evaporation rate coefficients $\gamma_{(i+j) \to i,j}$ are derived from $\Delta G$ of clusters and the corresponding
collision coefficients based on the detailed balance assumption (Mcgrath et al., 2012):

$$\gamma_{(i+j) \to i,j} = \beta_{i,j} \frac{P_{ref}}{k_B T} \exp\left(\frac{\Delta G_{i+j} - \Delta G_i - \Delta G_j}{k_B T}\right) \qquad (4)$$

where $P_{ref}$ is the reference pressure (1 atm) at which the Gibbs free energies were determined, and $\Delta G_i$ is the Gibbs formation
free energy of the formation of cluster $i$ from the corresponding monomers.

**3 Results**

**3.1 Cluster conformational analysis**

The obtained most stable structures of $(IA)_x \cdot (MSA)_y$ ($0 \le x \le 6$, $0 \le y \le 3$, $1 < x + y \le 6$) clusters are presented in Fig. S1 and the
corresponding Cartesian coordinates are collected in Table S7 in the supplement. To investigate the intermolecular bonding
potential of IA and MSA, the electrostatic potential (ESP) was calculated to analyse their potential interaction sites.

105        As shown in Fig. 1a, IA has positive ESPs (red region) surrounding its -OH group, with a maximum value of +59.04 kcal
mol$^{-1}$, making the -OH group an effective hydrogen bond (HB) donor. And IA's two terminal oxygens with negative ESPs (-
29.09 kcal mol$^{-1}$ and -29.47 kcal mol$^{-1}$) can serve as HB acceptors. Similarly, the -OH group of MSA has the strongest
electrophilicity (ESP value of +63.86 kcal mol$^{-1}$) as the HB donor, while its terminal O atom has strong nucleophilicity as the
HB acceptor, due to its lone pair of electrons. In this case, IA and MSA can directly bind with each other via HBs. Moreover,
IA possesses positive charge localization (the so-called $\delta$-hole) with a maximal ESP value of +51.87 kcal mol$^{-1}$ at the end of
the iodine atom along the O-I direction. This electron deficient region tends to attract the electron-rich oxygen atom of the
MSA to form the halogen bonds (XB) O-I···O (green band line in Fig. 1a). From the skeletal formula presented in Fig 1. (a),
we can know that the formed $(IA)_1 \cdot (MSA)_1$ cluster is stabilized by both HB and XB. The similar situation has also been found
in the larger IA-MSA clusters in Fig. S1.





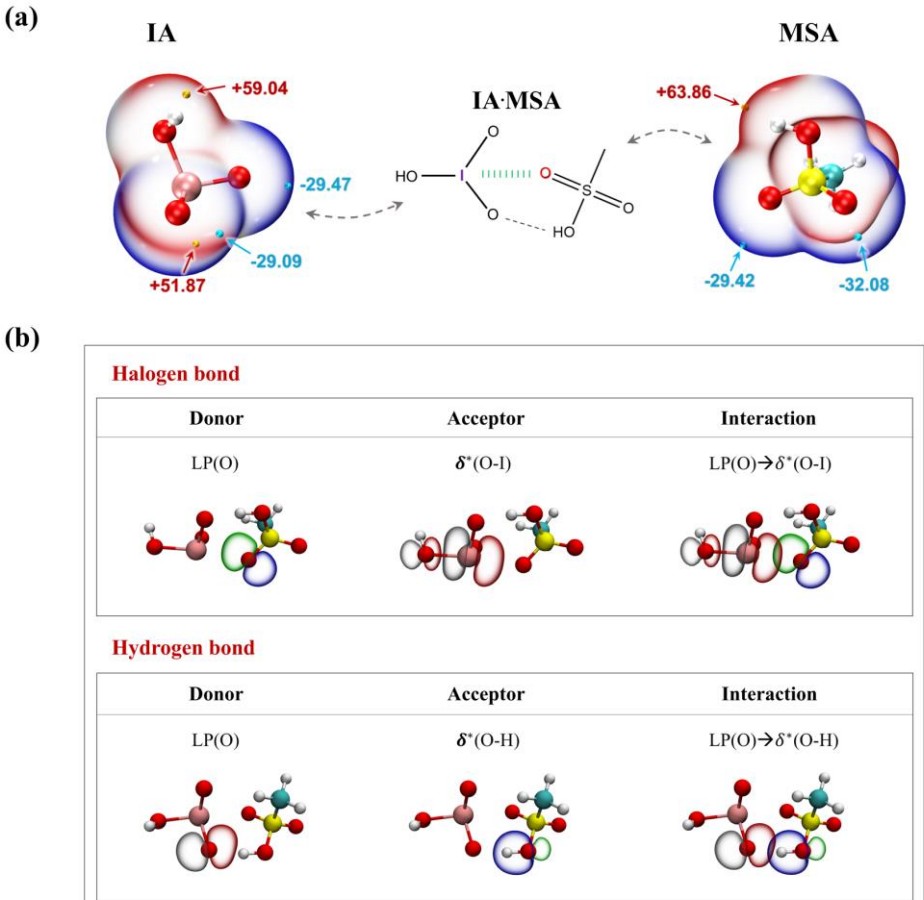


**Figure 1. (a)** The ESP-mapped molecular vdW surfaces of iodic acid (IA) and methanesulfonic acid (MSA). The pink, red, yellow, cyan, and white spheres represent I, O, S, C and H atoms, respectively. The yellow and cyan dots indicate the positions of maximums and minimums of ESP in kcal mol$^{-1}$, respectively. **(b)** The donor–acceptor NBOs involved in the $(IA)_1 \cdot (MSA)_1$ cluster. LP indicates the lone-pair orbitals, and $\delta^*$ indicates the antibonding orbitals.

Additionally, as illustrated in Fig. 1b, the natural bond orbital (NBO) analysis was performed to reveal the bonding nature of IA and  MSA. For the formed O-I$\cdots$O halogen bond, the lone-pair orbitals LP(O) in the terminal oxygen atom of MSA acts as an electron donor, while the antibonding orbital $\delta^*$(O-I) in IA is the electron acceptor. Essentially, halogen bonding originates from the interactions between LP(O) and $\delta^*$(O-I) orbitals, accompanied by intermolecular charge transferring from LP(O) to $\delta^*$(O-I). In the case of the O-H$\cdots$O hydrogen bond, the LP(O) of IA serves as the donor orbital and $\delta^*$(O-H) of MSA

is the acceptor orbital, and the charge shifts from LP(O) to $\delta^*$(O-H). The ESP and NBO results indicate that IA and MSA are capable of forming both HB and XB and have the potential to form stable clusters.

To quantify the bonding strength of HBs or XBs within the studied IA-MSA clusters (Fig. S1 in ESI†), the bonding properties including electron density $\rho(r)$, Laplacian electron density $\nabla^2\rho(r)$ and energy density $H(r)$ at the bond critical points





(BCPs), are calculated based on AIM methodology (Becke, 2007; Lane et al., 2013) and collected in Table S1. For O-I··O XBs, the $\rho(r)$, $\nabla^2\rho(r)$ and $H(r)$ values at the BCPs are in the ranges of 0.0143−0.0849, 0.0409−0.1589, and -0.0265−0.0019 a.u., respectively. As to O-H··O HBs, the $\rho(r)$, $\nabla^2\rho(r)$ and $H(r)$ values at the BCPs are in the ranges of 0.0178−0.0796, 0.0615−0.1141, and -0.0032−-0.0332 a.u., respectively. The electron density $\rho(r)$ is generally positively correlated with the bond strength, and the $\rho(r)$ values (Table S1 in ESI†) are well within the specified $\rho(r)$ range of HBs (0.002−0.040 a.u.) (Grabowski, 2004; Koch and Popelier, 1995) indicating that all the O-H··O non-covalent interactions are indeed HBs.

Moreover, according to the classification of HBs (Rozas et al., 2000), all the HBs formed within IA-MSA clusters are medium HBs (12.0 < $E$ (interaction energy) < 24.0 kcal mol$^{-1}$) with $\nabla^2\rho(r) > 0$ and $H(r) < 0$. Overall, the conformational analysis suggests that MSA can stabilize IA clusters (Fig. S1) by forming relative strong non-covalent interactions such as HBs and XBs, and thus has the potential to form stable clusters with IA.

## 3.2 Cluster stability analysis

To evaluate the thermodynamic stability of formed IA-MSA clusters, the Gibbs formation free energy ($\Delta G$, kcal mol$^{-1}$) and evaporation rate coefficient ($\gamma$, s$^{-1}$) of each studied $(IA)_x \cdot (MSA)_y$ ($0 \leq x \leq 6$, $0 \leq y \leq 3$, $1 < x + y \leq 6$) clusters at $T$ = 218 K−298 K ranging from boundary layer to troposphere (Williamson et al., 2019) and $p$ = 1 atm were calculated by Eq. (1) and Eq. (4) and shown in Table S2.

As shown in Fig. 2a, the $\Delta G$s of IA-MSA clusters at 278 K decrease with the increasing of cluster size, indicating that the cluster growth process is energetically favourable. The $\Delta G$ of the $(IA)_x \cdot (MSA)_1$ ($x$ = 1−5) clusters, are 7.71−15.67 kcal mol$^{-1}$ lower than that of the corresponding $(IA)_x$ clusters, indicating that pure-IA clusters could potentially grow up by binding with MSA. Moreover, the corresponding total evaporation rate coefficients ($\sum\gamma$, s$^{-1}$) of clusters were calculated by Eq. (4) and presented in Fig. 2b. In general, a lower $\sum\gamma$ value indicates greater cluster stability. As shown in Fig. 2b, the $\sum\gamma$ of larger

clusters, $(IA)_{4-6}$ and $(IA)_{3-4} \cdot (MSA)_2$, are significantly lower than those of the corresponding initial small-sized clusters, indicating that the stability increases during the cluster growth. Considering the competition between collision and evaporation during the clustering process, the ratio of collision frequencies versus total evaporation rate coefficients ($\beta C/\sum\gamma$) was calculated as the probability of cluster growth (Fig. 2c). Among these clusters, the largest $(IA)_4 \cdot (MSA)_2$ and $(IA)_6$ clusters have sufficient stability against decomposition ($\beta C/\sum\gamma > 1$) and thus continue to grow. As a result, the fluxes for clusters with larger size than

$(IA)_4 \cdot (MSA)_2$ and $(IA)_6$ are counted in the cluster formation rate $J$.





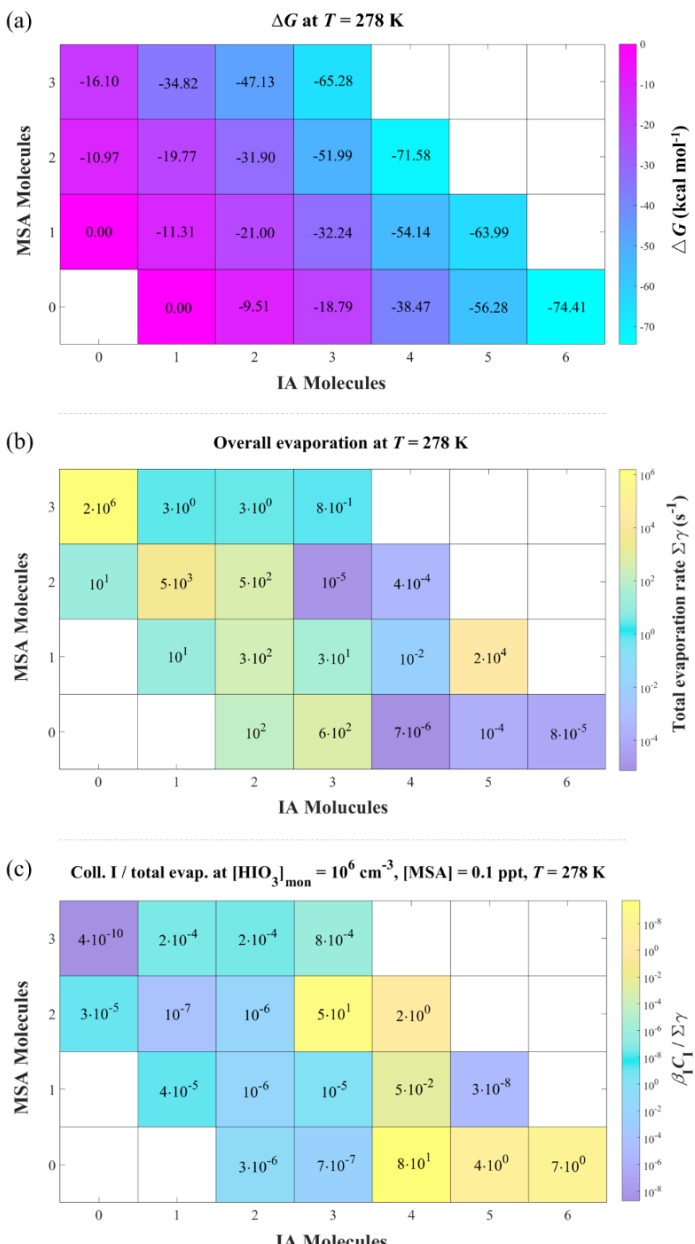

**Figure 2. (a)** Gibbs formation free energy ($\Delta G$, kcal mol$^{-1}$) of the $(IA)_x \cdot (MSA)_y$ ($0 \le x \le 6$, $0 \le y \le 3$, $1 < x + y \le 6$) clusters calculated at the RI-CC2/aug-cc-pV(T+d)Z(-PP)//$\omega$B97X-D/6-311++G(3df,3pd) + aug-cc-pVTZ(-PP) level of theory, $T = 278$ K, $p = 1$ atm. **(b)** the total evaporation rate coefficients ($\sum \gamma$, s$^{-1}$) and **(c)** the ratios of IA monomer collision frequencies versus total evaporation rate coefficients ($\beta C / \sum \gamma$) of the corresponding clusters.





### 3.3 Cluster formation rates

To comprehensively explore the effect of MSA on IA cluster formation kinetically, the IA-MSA-based cluster formation rate $J$ (cm$^{-3}$ s$^{-1}$) was simulated under different atmospheric conditions using ACDC. Herein, we first explored the changes of $J$ after the intervention of different concentrations of MSA ([MSA]), using the pure-IA system as a reference. Based on the field measurement (Sipila et al., 2016), [IA] in the ACDC simulation is set to be in the range of $10^6-10^8$ molecules cm$^{-3}$, [MSA] = $10^6$, $10^7$, and $10^8$ molecules cm$^{-3}$ (Chen et al., 2018; Berresheim et al., 2002; Davis et al., 1998). As to the setting of condensation sink coefficient (CS), the different CS ($1.0\times10^{-4} \sim 2.6\times10^{-3}$ s$^{-1}$) have a minor impact on the cluster formation rate (Fig. S4), especially in the case of relatively higher [IA] and [MSA]. Hence, the CS is chosen as a typical coastal value ($2.0\times10^{-3}$ s$^{-1}$) (Dal Maso et al., 2002).

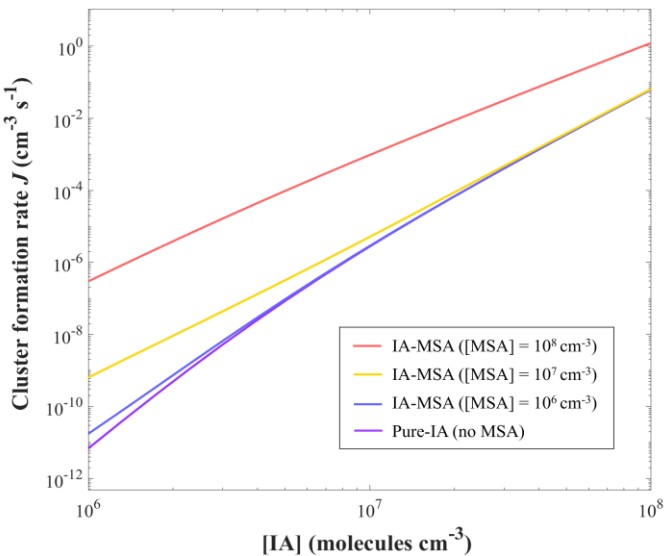

**Figure 3.** Simulated cluster formation rates $J$ (cm$^{-3}$ s$^{-1}$) as a function of iodic acid concentration [IA], with different concentrations of methanesulfonic acid [MSA] of $10^6$ (blue), $10^7$ (yellow), $10^8$ (red) and 0 molecules cm$^{-3}$ (purple, pure-IA), at $T$ = 278 K, CS = $2.0\times10^{-3}$ s$^{-1}$.

As shown in Fig. 3, the $J$ of IA-MSA system with varying [MSA] (red, yellow, and blue lines) are all higher than that of the pure-IA system (purple line). Particularly, at a lower [IA] of $10^6$ molecules cm$^{-3}$, the involvement of MSA results in a greater boost on $J$. Even at a median of [MSA] ($10^7$ molecules cm$^{-3}$), the $J$ of the pure-IA system can be improved by nearly two orders of magnitude in this case. Briefly, MSA can promote $J$ of IA clusters to a higher level, which may help explain the rapid formation of IA-involved particles in marine NPF.

To quantify such enhancement of the MSA on $J$, here we defined an enhancement strength $R$ as the following Eq. (5):

$$R = \frac{J_{\text{IA-MSA}}}{J_{\text{pure-IA}}} = \frac{J([IA] = x, [MSA] = y)}{J([IA] = x)} \tag{5}$$




where $J_{\text{IA-MSA}}$ and $J_{\text{pure-IA}}$ indicates the $J$ of IA-MSA and pure-IA nucleating system, respectively. $x$ and $y$ are the atmospheric concentrations of IA and MSA, respectively.

During nucleating processes, variations in ambient conditions (precursor concentration and temperature) can affect $J_{\text{IA-MSA}}$ and $J_{\text{pure-IA}}$ as well as the $R$ of MSA. Herein, the simulations were performed in a wide range of atmospheric temperatures ($T = 218$ K$-298$ K) and common concentrations of IA ($10^6-10^8$ molecules cm$^{-3}$) and MSA ($10^6-10^8$ molecules cm$^{-3}$).

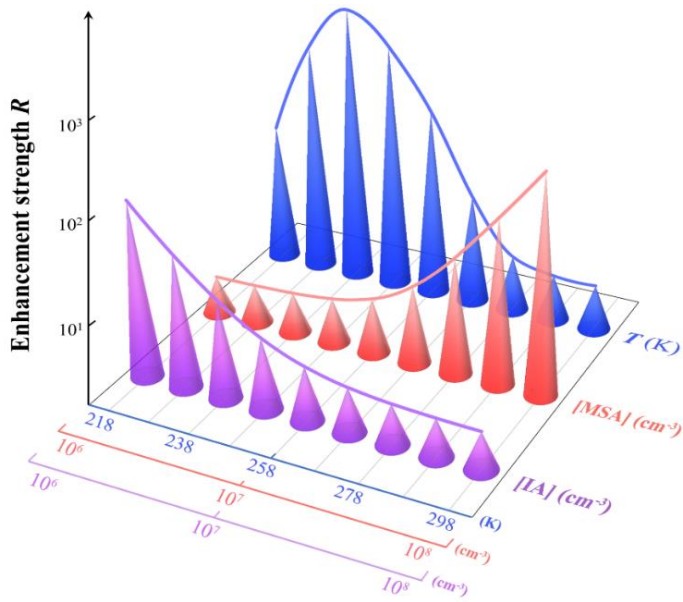


**Figure 4**. Enhancement strength $R$ of MSA on cluster formation rates under different atmospheric conditions: $T = 218-298$ K, [MSA] $= 10^6-10^8$ molecules cm$^{-3}$, [IA] $= 10^6-10^8$ molecules cm$^{-3}$, and CS $= 2.0 \times 10^{-3}$ s$^{-1}$.

As shown in Fig. 4, the enhancement strength $R$ of MSA decreases with the increasing of [IA] ($10^6 \rightarrow 10^8$ molecules cm$^{-3}$), under the condition of $T = 278$ K and [MSA] $= 10^7$ molecules cm$^{-3}$ (purple line). This is because the contribution of pure-IA

clusters to nucleation becomes higher with the increasing of [IA], thereby diminishing that of IA-MSA clusters (smaller $R$). Conversely at lower [IA] ($10^6$ molecules cm$^{-3}$), the $R$ of MSA on $J$ could reach 98-fold, nearly two orders of magnitude, even when [MSA] is only at a median value ($10^7$ molecules cm$^{-3}$). The $R$ increases with the increasing of [MSA] ($10^6 \rightarrow 10^8$ molecules cm$^{-3}$) (orange line) due to more IA-MSA clusters formed, resulting in $R_{\max} = 347$ at [MSA] $= 10^8$ and [IA] $= 10^7$ molecules cm$^{-3}$. Interestingly, as the temperature decreases from 298 K to 218 K (blue line), $R$ first increases (298 K $\rightarrow$ 238

K) and then decreases (238 K $\rightarrow$ 218 K). During the temperature range from 298 K to 238 K, the decrease in temperature diminishes cluster evaporation (Eq. (4)), which in turn promotes IA-MSA cluster formation and leads to an increase in $R$. When the temperature is very low, between 218 and 238 K, the effect of cluster evaporation is almost negligible, and the nucleation process is primarily limited by collisions between clusters or molecules, namely the kinetic limit process. In this case, the lower $T$ reduces the collision rate and thus results in a decrease in $R$. As a result of the above analysis, the effect of





the IA-MSA system on the nucleation process varies with the [IA], [MSA] and $T$, and is particularly important in regions with lower $T$, sparse IA and rich MSA.

### 3.4 Cluster growth pathways

According to the analysis above, MSA can stabilize IA clusters, thereby enhancing cluster formation rate. However, the
mechanism of how MSA and IA jointly contribute to cluster formation is still unclear. Thus, the detailed cluster growth pathways were tracked by ACDC and shown in Fig. 5a.



**Figure 5. (a)** Main cluster growth pathway of IA-MSA nucleating system at $T = 258-298$ K, CS $= 2.0 \times 10^{-3}$ s$^{-1}$, [IA] $= 10^7$ and [MSA] $= 10^7$ molecules cm$^{-3}$. The black and orange arrows refer to the pathways of colliding with IA and MSA, respectively, where the dashed arrows indicate the evaporation of MSA. **(b)** Branch ratio of IA-MSA (orange pie) and pure-IA (purple pie) growth pathway in different regions with different [IA]. The map is from © Google Maps (https://www.google.com/maps).

The main clustering pathways can be divided into two types: i) IA self-nucleation and ii) IA-MSA cluster formation. The studied clusters that did not appear in the cluster growth pathway are mainly due to their low stability. For the IA self-nucleation pathway, cluster growth proceeds mainly via the collisional binding of IA monomers ((IA)$_{1\to2\to3\to4\to5\to6}$), which is consistent with the reported pure-IA nucleation mechanism (Rong et al., 2020; Sipila et al., 2016). For the IA-MSA pathway, it starts from the heterodimer (IA)$_1\cdot$(MSA)$_1$, and then grows primarily through IA addition, resulting in the (IA)$_4\cdot$(MSA)$_2$ clusters with sufficient stability to grow out of the simulated system. The results suggest that MSA can directly participate in the IA-involved nucleation by forming stable IA-MSA clusters. And this findings can provide theoretical evidence to explain some MSA and IA detected in the particle phase (Beck et al., 2020) through the IA-MSA nucleating pathway.

To gain more insight into the importance of the IA-MSA mechanism, the specific contribution of these cluster formation pathways needs to be further quantified. The contribution of the IA-MSA pathways varies with the ambient conditions in different areas. Thus, as shown in Fig. 5b, the branch ratios of IA-MSA and IA self-nucleation pathways were calculated based on field data (He et al., 2021) for the mid-latitude coastal areas, e.g., Helsinki (60°12′N, 24°58′E), Mace Head (53°19′N, 9°53′W), Beijing (39°94′N, 116°30′E), and Réunion (21.2°S, 55.7°E), and high-latitude polar regions such as Greenland (81°36′N, 16°40′W), Ny-Ålesund (78°55′N, 11°56′ E), Aboa (73°03′S, 13°25′W) and Neumayer (73°58′S, 8°23′W). In the ACDC simulations, the atmospheric temperature and [IA] were set to be the average observation values of each location shown in Fig. 5b. The corresponding $\Delta G$s at different temperatures were calculated in Table S3. Due to the lack of field data for MSA at multiple sites, a median of [MSA] ($10^7$ molecules cm$^{-3}$) was chosen in this study. For near-polar regions at high latitudes, Aboa, Neumayer, Greenland, and Ny-Ålesund, the CS was set to be $1 \times 10^{-4}$ s$^{-1}$ (Baccarini et al., 2020). For mid-latitude coastal regions, Mace Head, Helsinki and Réunion, the CS was set to be $2 \times 10^{-3}$ s$^{-1}$ (Dal Maso et al., 2002). As to Beijing with pollution, the CS was set to be $2 \times 10^{-2}$ s$^{-1}$ (Yao et al., 2018).

As illustrated in Fig. 5b, the IA-MSA pathways contribute more to nucleation in polar regions than that in mid-latitude coastal regions, owing to the lower temperatures. Specifically, at mid-latitude regions with higher [IA] ($10^8$ molecules cm$^{-3}$) such as Mace Head, the contribution of IA-MSA pathway is minor and the NPF process is dominated by IA self-nucleation, which is consistent with the previous findings (Sipila et al., 2016). While in regions with relatively lower [IA], such as Helsinki ([IA] $= 10^7$ molecules cm$^{-3}$), Beijing and Réunion ([IA] $= 3 \times 10^6$ molecules cm$^{-3}$), the contribution of IA-MSA pathways is 14% (Helsinki), 27% (Beijing), and 62% (Réunion), respectively. For cold near-polar regions, such as Aboa, Neumayer, Greenland, and Ny-Ålesund, the IA-MSA pathway contributes more than 90% to cluster formation. Furthermore, when comparing Neumayer and Helsinki with same [IA], the decrease of temperature leads to a significant increase in IA-MSA contribution (14% to 91%). Comparing Neumayer and Aboa with the same temperature, [IA] is reduced by 70% ($10^7$ to $3 \times 10^6$ molecules





cm$^{-3}$), resulting in 8% improvement in IA-MSA contribution. Overall, the contribution of IA-MSA nucleation pathway is mainly influenced by temperature, followed by the abundance of IA, indicating that IA-MSA nucleation is especially critical for cold polar regions with lower [IA]. It should be noted that the above simulation results in Fig. 5b are obtained at a fixed [MSA] of a median value ($1 \times 10^7$ molecules cm$^{-3}$) due to the lack of available systematic distribution of MSA across regions.

If the simulations were performed at different [MSA], the conclusions would be different. For example, during the Arctic Ocean 2018 expedition (Baccarini et al., 2020), the observed [MSA] is significantly lower than [IA]. Under such atmospheric conditions ($T = 268$ K, CS = $1\times10^{-4}$ s$^{-1}$, [IA] = $10^5 \sim 10^6$ and [MSA] = $10^5$ molecules cm$^{-3}$), the simulation results show that the contribution of IA-MSA clustering pathway is only 1%$-$16%. This implies that in environments with lower [MSA], IA nucleation still dominates, which is also consistent with the findings of Baccarini et al, (2020). In general, the IA-MSA

mechanism contributed more prominently in cold polar regions especially with higher [MSA] and lower [IA].

**3.5 Comparison with field measurement**

For polar regions, the recent field study (Beck et al., 2020) have shown that [IA] is lower ($10^6-10^7$ molecules cm$^{-3}$) in the spring of Ny-Ålesund, but significant NPF is observed. Another distinctive feature of this period is the relatively higher [MSA] ($10^6-3.3\times10^7$ molecules cm$^{-3}$) in the atmosphere. In this case, IA and MSA are likely to nucleate together, therefore, the $J$ of IA-MSA system was simulated under the field conditions observed ($T = 268$ K, [MSA] = $10^6-3.3\times10^7$, [IA] = $10^6-10^7$

molecules cm$^{-3}$, CS = $4\times10^{-4}$ s$^{-1}$).

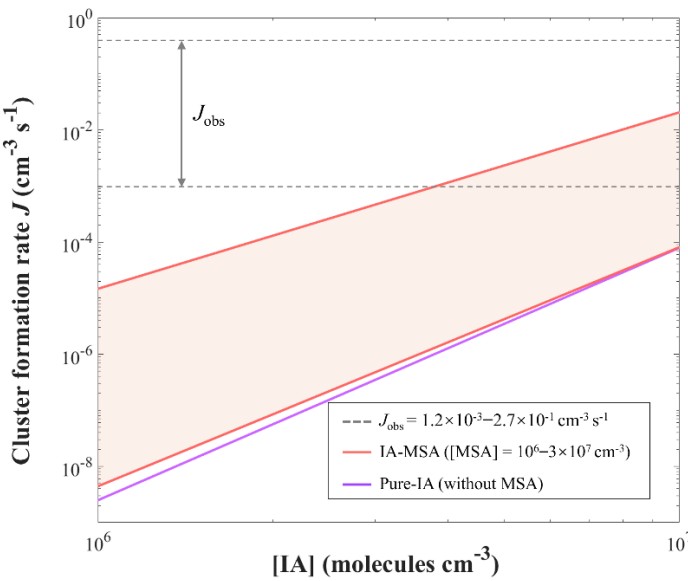

**Figure 6**. The simulated cluster formation rate $J$ in Ny-Ålesund ($T = 268$ K, CS = $4\times10^{-4}$ s$^{-1}$, [IA] = $10^6-10^7$, [MSA] = $10^6-3.3\times10^7$ molecules cm$^{-3}$. The purple line indicates the $J$ of pure-IA system. The red lines refer to the $J$ of IA-MSA system

at varying [MSA] ($10^6-3.3\times10^7$ molecules cm$^{-3}$). $J_{obs}$ (cm$^{-3}$ s$^{-1}$) is the observed cluster formation rate in Ny-Ålesund.



Fig. 6 presents a comparison of the simulated $J$ by ACDC with the observed $J_{obs}$ ($10^{-3}$–$2.7 \times 10^{-1}$ cm$^{-3}$ s$^{-1}$) of Ny-Ålesund (Beck et al., 2020). The results indicate that IA self-nucleation (purple line) alone is not sufficient to explain the $J_{obs}$, while the higher $J$ of IA-MSA nucleation system (red area) is in good agreement with $J_{obs}$. Thus, compared with the IA self-nucleation, the IA-MSA nucleation mechanism potentially plays a significant role in the cold Ny-Ålesund with relatively rich MSA and sparse

IA. However, there is still a partially unmatched $J_{obs}$, which is supposed to be contributed by the SA- ammonia (NH$_3$) ion-induced nucleation proposed by Beck et al. (2020). And this ion-induced nucleation mechanism is not considered in the present study.

In fact, in the real atmosphere, besides IA and MSA, other components such as iodous acid (HIO$_2$), iodine oxides (I$_2$O$_4$ and I$_2$O$_5$), and SA, and NH$_3$, etc., can also potentially contribute to marine nucleation. Such a multi-component nucleation study

will be continued to be carried out gradually in future work to construct a more comprehensive marine multi-component nucleation model.

## 4 Atmospheric significance and conclusion

The present work systematically investigates the joint nucleation mechanisms of two critical marine nucleation precursors, methanesulfonic acid (MSA) and iodic acid (IA), using the quantum chemical approach and Atmospheric Cluster Dynamics

Code (ACDC). The results suggest that MSA can stabilize IA clusters structurally by forming hydrogen and halogen bonds, and the formed IA-MSA clusters are thermodynamically stable enough to grow further. Additionally, kinetic simulations by ACDC indicate that MSA can significantly enhance the formation rate of IA clusters, particularly at higher [MSA], lower [IA] and $T$. The corresponding IA-MSA nucleating mechanism can be described by two distinct pathways: i) pure-IA cluster formation and ii) IA-MSA cluster formation, indicating that IA and MSA can jointly nucleate. Moreover, the IA-MSA pathway

contributes more than 90% to cluster formation in the high-latitude polar regions with higher [MSA] and lower [IA], and 14%, 27%, and 62% to cluster formation of the mid-latitude coastal regions, Helsinki, Beijing and Réunion, respectively. Furthermore, compared with the field measurement in Ny-Ålesund with higher [MSA], the IA self-nucleation is insufficient to account for the observed cluster formation rates, whereas the IA-MSA system provides a better fit. These results suggest that the joint nucleation mechanisms of IA and MSA plays a significant role in marine NPF, especially in polar regions with

rich MSA and sparse IA.

The current study provides evidence that IA and MSA can jointly nucleate at the molecular level and the IA-MSA joint nucleation is more efficient than the reported IA self-nucleation, which can help to explain the intensive marine NPF events. More broadly, this finding helps to construct a more comprehensive marine multi-component nucleation model.






*Competing interests*. The authors declare that they have no conflict of interest.

*Data availability*. The data in this article are available from the corresponding author upon request (zhangxiuhui@bit.edu.cn)

*Author contributions*. AN: Data curation, Formal analysis, Investigation, Visualization, Writing – original draft preparation; LL: Validation, Visualization. LJ: Validation, Writing review & editing. XHZ: Supervision, Conceptualization, Writing - review & editing, Formal analysis, Data curation.

*Acknowledgements*. We acknowledge the National Supercomputing Centre in Shenzhen for providing the computational resources and Turbomole program.

*Financial support*. This work was supported by the National Natural Science Foundation of China (21976015). L. L. thanks the China Postdoctoral Science Foundation (2020M680013).



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
