# Peer review of "Molecular-level evidence for marine aerosol nucleation of iodic acid and methanesulfonic acid"

_Atmospheric Chemistry and Physics, 2021_

## Author Comment (AC1)

**Responses to Referee #1's comments**

Thanks very much for taking your time to review this manuscript. We really appreciate all your valuable and helpful comments on our manuscript "Molecular-level evidence for marine aerosol nucleation of iodic acid and methanesulfonic acid" (MS No.: acp-2021-595). We have revised the manuscript carefully according to reviewer's comments. The point-to-point responses to the Referee #1's comments are summarized below:

**General Comments:**

Particle nucleation events have been repeatedly observed in marine environments and are associated with large increases in the concentration of particles smaller than 20 nm. While atmospheric observations provide the definitive evidence on which compounds are essential for this process, computational methods have the advantage of studying simple binary or ternary systems and revealing important interactions. Ning et al. investigated the nucleation mechanisms of iodic acid (IA) and methane sulfonic acid (MSA) using high level quantum chemical calculations combined with the Atmospheric Clusters Dynamic Code (ACDC). They proved that MSA can participate in the early nucleation steps with $HIO_3$ molecules, at least from a molecular dynamic point of view. They further show that the MSA enhancement over the $HIO_3$ system is dependent on the $HIO_3$ concentration and the temperature. The paper is well written and presents new insights into the marine nucleation mechanism. Therefore, I recommend the publication of this study in ACP after considering the comments listed below.

**Response:** We would like to thank the reviewer for the positive and valuable comments, and we have revised our manuscript accordingly.
* * *
**Comment 1:** The authors have put a big emphasis on comparing their results to atmospheric observations, which is invalid in some cases and has weakened this study. For example, Figure 5b assumes that MSA concentration is equal to $1 \times 10^7$ molecules/cm$^3$ in all presented sites, clearly overestimating the MSA concentration in many locations. Additionally, the comparison to Beck (Beck et al., 2021)et al. (2020) shown in Figure 6 does not give additional merit to the proposed MSA-IA mechanism, especially that the authors are aware that sulfuric acid (SA) and ammonia seem to play a significant role at this site and that IA and SA could have a synergetic

role (Rong et al., 2020). It is recommended to put less emphasis on this comparison and instead focus on the results of the simulations, for example, moving figure S5 or S6 from the supplementary to the main text.

**Response:** We appreciate this constructive suggestion. As suggested by the reviewer (comment 1 and comment 19), the original Figure 5b has been moved to the supplement in order to weaken the comparison with the field observation. In its place, the redrawn Figure 5b in the revised manuscript presents the contribution of the IA-MSA clustering pathway at different [MSA] ($10^6$ – $10^7$ molecules cm$^{-3}$) and [IA] ($10^6$ – $10^8$ molecules cm$^{-3}$) in a similar form to Rong et al. (2020)'s Figure 3b. To further put less emphasis on comparison to Beck et al. (2020), Figure 6 has been replaced with Figure S5 in the main text according to the reviewer's suggestion.
* * *
**Comment 2:** The authors are encouraged to discuss the reasons behind the discrepancy in the formation rates presented here and in a previous study. The same group have reported that the formation rates of the pure IA system at [IA] of $1\times10^8$ cm$^{-3}$ with a temperature of 278K and $2\times10^{-3}$ s$^{-1}$ CS is below $1\times10^{-5}$ cm$^{-3}$ s$^{-1}$ (Rong et al., 2020), while the formation rates presented in Figure 3 of this study at similar conditions is higher than $1\times10^{-2}$ cm$^{-3}$ s$^{-1}$.

**Response:** Thanks for your suggestions. The discrepancy in the formation rates is attributed to the fact that the cluster structure and thermodynamic properties were calculated at a different level of theory in the present work. In the previous study (Rong et al., 2020), the double-ζ basis set (aug-cc-pVDZ-PP) was employed for iodine atom. To get more accurate results, the larger triple-ζ basis set (aug-cc-pVTZ-PP) was used for iodine in the quantum chemical calculations of the present manuscript. In that case, the simulated $J$ by ACDC based on the different $\Delta G$s of the clusters obtained by quantum chemical calculations will differ because of the sensitivity of cluster evaporation to $\Delta G$.
* * *
**Comment 3:** The authors should also further discuss the limits of this study, causing 'discrepancies' with results reported in the literature. A very brief explanation is currently given in lines 273-274, but it is not sufficient. Optimally, the reader would understand the limits of this study compared to chamber or atmospheric measurements at an early stage of the manuscript. For example, the authors should discuss the difference between this study and that of He et al. (2021), resulting in different formation rates for the pure IA system, or that MSA is

never present in the atmosphere without SA or that the MSA clusters are expected to be stabilized by water in the atmosphere (Chen et al., 2020).

**Response:** This is a very pertinent point – thank you for bringing it up. According to the reviewer's suggestion, we have added a description of the limitations of the IA-MSA nucleation mechanism in lines 106-111 of the revised manuscript as follows:

"In addition, for nucleation processes driven by iodine-containing components, the significant impact of $HIO_2$ and iodine oxides ($I_2O_4$ and $I_2O_5$) needs to be considered (He et al., 2021). The present study focuses more on the nucleation mechanism of MSA and the essential driver IA. In the real marine atmosphere, other nucleation precursors, such as SA, $NH_3$, amine, etc., may also affect the nucleation process. Particularly with SA, because MSA and SA coexist in the air and both are formed during the oxidation of DMS in the marine atmosphere. The settings of the boundary conditions of the ACDC simulations are summarized in Table S5."
* * *
**Specific comments:**

**Comment 4.**

**Line 42:** Please add here the corrections He et al. (2021) made on the Sipila et al. (2016) proposed IA self-nucleation mechanism.

**Response:** According to the reviewer's helpful suggestion, the corrections He et al. (2021) made on the Sipila et al. (2016) was added in lines 42-44 of the revised manuscript as follows:

"…the coastal NPF is primarily driven by subsequential addition of IA and involves the participation of $I_2O_5$. More recently, He et al. (2021) demonstrated experimentally that, in addition to IA and $I_2O_5$, iodous acid ($HIO_2$) and $I_2O_4$ are also involved in the cluster formation process, with $HIO_2$ playing a key role in the stabilization of neutral IA clusters."
* * *
**Comment 5.**

**Line 45:** Beck et al. (2020) did not measure MSA and IA in the particle phase but in clusters using a CI-API-TOF (which could be gaseous). Thus, the sentence in its current form is misleading.

**Response:** Thanks for the reviewer's reminding. "…in the particle phase" has been corrected as "…in the clusters" according to the study of Beck et al. (2020).
* * *
**Comment 6.**

**Line 83:** There is no footnote for the electronic supplementary information (ESI). Please remove the symbol after 'ESI'. (Also in lines 127 and 133).

**Response:** According to the reviewer's suggestion, all the symbols after 'ESI' have been removed from the revised manuscript.
* * *
**Comment 7.**

**Line 84:** Please add more information on the ACDC simulations. For example, that the simulations do not include the effect of water or charge.

**Response:** As suggested by the reviewer, more information on ACDC simulations has been added in lines 103–106 of the revised manuscript as follows:

"In the present study, the ACDC simulations only modelled the neutral cluster formation process and did not consider the charge, nor the effect of water. Since IA is weakly bound to water, it is less inclined to exist as hydration of IA in tropospheric conditions (Khanniche et al., 2016). Meanwhile, the nucleation efficiency of MSA and water is very low (Arquero et al., 2017). Thus, the effect of water on the conclusion is limited."
* * *
**Comment 8.**

**Line 88:** What does the $J$ in equation (2) stand for? It is misleading to have $J$ here because the reader would think that it refers to formation rate, and the formation rate is not equal to dc/dt.

**Response:** Thanks for the reviewer's professional suggestion. $J$ does cause some misleading in equation (2) and has been removed from that equation.
* * *
**Comment 9.**

**Line 99:** Please refer to the ACDC boundary conditions presented in Table S5 in this section or somewhere else in the text.

**Response:** According to the reviewer's suggestion, the ACDC boundary condition has been referred to the Table S5 in line 111 of the revised manuscript. The added content is: "The

settings of the boundary conditions of the ACDC simulations are summarized in Table S5."
* * *
**Comment 10.**

**Line 113:** Please replace 'the' by 'a' in the sentence: The similar situation...

**Response:** As suggested by the reviewer, 'the' has been replaced with 'a' in the similar situation of the manuscript.
* * *
**Comment 11.**

**Line 144:** Table S2 contains information about the Gibbs formation free energy only and does not include evaporation rates. Evaporation rates are presented in Table S4 and only at one temperature. This should be clarified.

**Response:** Thanks for the reviewer's suggestion. The contents in Table 2 and Table 4 were clarified in lines 156 and 161 of the revised manuscript separately.
* * *
**Comment 12.**

**Line 149:** Refer to Table S4 after referring to Fig. 2b.

**Response:** According to the reviewer's suggestion, Table S4 has been referred after referring to Fig. 2b in the revised manuscript.
* * *
**Comment 13.**

**Line 155:** The supplement also shows similar figures to Fig. 2 but at 298 K (Fig. S2) and 258 K (Fig. S3). Please refer to these figures in the main text or delete them.

**Response:** According to the reviewer's suggestion, Fig. S2 and Fig. S3 have been referred in the main text.
* * *
**Comment 14.**

**Line 171:** Should this be referring to the coagulation sink instead?

**Response:** Thanks for the reviewer's valuable suggestion. Coagulation sink is indeed an important treatment. Considering that a cluster size dependent coagulation sink coefficient has

no important effect on steady-state cluster concentrations (McGrath et al., 2012), the constant condensation sink coefficients were chosen in the ACDC simulations of the present study.
* * *
**Comment 15.**

**Line 191:** Please adjust the caption of Fig. 4 to include the MSA concentration in the purple cones, the IA concentration in the red cones and the IA and MSA concentration in the blue cones.

**Response:** According to the reviewer's suggestion, the description of the color of cones in the caption of Figure 4 has been added to the revised manuscript.
* * *
**Comment 16.**

**Line 193:** Also refer to Table S6 here.

**Response:** Table 6 has been referred in line 205 of the revised manuscript as follows: "The specific $R$ values were summarized in Table S6."
* * *
**Comment 17.**

**Line 193:** Please refer to and discuss Figure S5 while presenting the temperature effect.

**Response:** According to the reviewer's suggestion, Figure S5 has been referred and the corresponding discussion to were added in the lines 243-245 of revised manuscript.
* * *
**Comment 18.**

**Line 224:** Beck et al. (2020) did not show MSA-IA clusters and did not measure these exclusively in the particle phase (see comment on Line 45), so this reference cannot be used here to support your conclusion here.

**Response:** According to the reviewer's suggestion, the citation of Beck et al. (2020) has been removed from the revised manuscript.
* * *
**Comment 19.**

**Lines 225-255:** As the authors mention, the analysis shown in this section is highly dependent

on the chosen MSA concentration for the simulations. An average MSA concentration of $1\times10^7$ molecules cm$^{-3}$ is an overestimate for MSA measured in most of the cites sites. Thus, I suggest that the analysis is repeated with a more reasonable concentration or the reference to locations is omitted, and a figure similar to Rong et al. (2020)'s Figure 3b is presented instead (it could also be presented as a stacked bar graph with different temperatures listed next to each other). Otherwise, Figure 5b can be moved to the supplement, and less emphasis on it is given in the main text.

**Response:** According to the reviewer's valuable suggestion, the mentioned analysis has been repeated at a more reasonable concentration of MSA ($2.5\times10^6$ molecules cm$^{-3}$) (Bork et al., 2014) and the resulting Figure 5b has been moved to the supplement (Fig. S5). In the revised manuscript, the modified Figure 5 presents the contribution of the IA-MSA clustering pathway at varying [MSA] ($10^6 - 10^7$ molecules cm$^{-3}$) and [IA] ($10^6 - 10^8$ molecules cm$^{-3}$), which is like Rong et al. (2020)'s Figure 3b. For your convenience, the modified Figure 5 in the revised manuscript is presented as following:

[Figure]

**Figure 5. (a)** Main cluster growth pathway of IA-MSA nucleating system at $T = 278$ K, CS = $2.0\times10^{-3}$ s$^{-1}$, [IA] = $10^7$ and [MSA] = $10^7$ molecules cm$^{-3}$. The black and orange arrows refer to the

pathways of colliding with IA and MSA, respectively, where the dashed arrows indicate the evaporation of MSA. **(b)** Branch ratio of IA-MSA (orange pie) and pure-IA (purple pie) growth pathway under varying $[MSA] = 10^6 - 10^7$ molecules cm$^{-3}$ and $[IA] = 10^6 - 10^8$ molecules cm$^{-3}$.

The corresponding statements of Figure 5b were added as follows:

"In the atmosphere, the distribution of IA and MSA varies by region, affecting the contribution of IA-MSA clustering pathways accordingly. Hence, the branch ratios of flux out through the IA-MSA path (orange pie) and pure-IA path (purple pie) at varying [MSA] ($10^6 - 10^7$ molecules cm$^{-3}$) and [IA] ($[IA] = 10^6 - 10^8$ molecules cm$^{-3}$) are presented in Fig. 5b to access the IA-MSA mechanism. As shown in Fig. 5b, the branch ratio of IA-MSA path and pure-IA is highly dependent on [MSA] and [IA]. At the condition of $T = 278$ K, CS $= 2.0 \times 10^{-3}$ s$^{-1}$ and $[IA] = 10^7$ molecules cm$^{-3}$, the contribution of IA-MSA path increases from 1% to 48% with the increasing of [MSA]. Additionally, given the uneven distribution of IA, the analysis was further carried out within the atmospherically relevant range of [IA] ($10^6 - 10^8$ molecules cm$^{-3}$). The results show that the contribution of IA-MSA path decreases from 97% to 4% with the increasing of [IA] ($10^6 \rightarrow 10^8$ molecules cm$^{-3}$). These findings indicate that the IA-MSA mechanism contributes more in colder regions with higher [MSA] and lower [IA]. Furthermore, the branch ratio was calculated based on field conditions (temperatures and [IA]) reported by He et al. (2021) and presented in Fig. S5. The results indicate that the IA-MSA mechanism does have stronger effects in polar regions than in mid-latitude coastal regions due to lower temperatures, which is also consistent with the above findings."

To further put less emphasis on comparison to Beck et al. (2020), Figure 6 in the revised manuscript has been replaced with Figure S5 according to the reviewer's suggestion.

[Figure]

**Figure 6.** The simulated cluster formation rate J (cm$^{-3}$ s$^{-1}$) of the IA-MSA system at different temperatures (a) 218, (b) 238, (c) 258, (d) 298 K, [IA] = $10^6$–$10^8$ molecules cm$^{-3}$, [MSA] = 0, $10^6$, $10^7$, $10^8$ molecules cm$^{-3}$, and CS = $2.0\times10^{-3}$ s$^{-1}$.

The corresponding statements of Figure 6 were added in the revised manuscript as follows:
"Most of the analysis above in the text was performed at 278 K. In fact, temperature has a strong influence on cluster formation, so it is necessary to further probe the impact of temperature on *J* systematically. Figure. 6 presents the simulated *J* at additional temperatures (218, 238, 258 and 298 K), [IA] = $10^6 - 10^8$ molecules cm$^{-3}$, [MSA] = $10^6$ (red line), $10^7$ (yellow line), $10^8$ (purple line) molecules cm$^{-3}$. At a relatively high *T* = 298 K (Fig. 6d), the improvement by the addition of MSA was not significant compared to the pure-IA system, except at higher [MSA] = $10^8$ molecules cm$^{-3}$ and relatively lower [IA]. At lower *T* = 258 K (Fig. 6c), the enhancement on *J* by MSA is stronger in all cases except at lowest [MSA] = $10^6$ molecules cm$^{-3}$. Moreover, such boost on *J* was further enhanced at 238 K (Fig. 6b). Lower concentrations of MSA ($10^6$ molecules cm$^{-3}$) also significantly promote the formation of IA clusters, mainly because the low temperature weakens the cluster evaporation."
* * *
**Comment 20.**

**Lines 256-276:** This section is dedicated for ACDC simulations at conditions of MSA, IA, temperature, and CS identical to those reported in Beck et al. (2020). However, the comparison to the measurements at Ny- Ålesund is not straightforward, as mentioned in the 1$^{st}$ general comment. Please discuss more the limitations or give less emphasis on this comparison.

**Response:** According to the reviewer's value suggestion, the comparison to the measurements at Ny- Ålesund (the original Figure 6) and the corresponding statement have been removed from the revised manuscript.
* * *
**Comment 21.**

**Lines 284-286:** This sentence must be rephrased to have a less strong statement because the analysis performed depends highly on the chosen MSA concentration.

**Response:** Thanks for the reviewer's constructive suggestion. The statement about the contribution of IA-MSA clustering pathways has been rephrased to a less strong form in lines

277-280 of the revised manuscript as follows:

"Moreover, the IA-MSA clustering pathway potentially contributes more in the colder polar regions, especially with higher [MSA] and lower [IA], than that of the mid-latitude coastal regions. The impact of the IA-MSA mechanism is highly dependent on the distribution of MSA and IA in the marine atmosphere."
* * *
**Comment 22.**

**Line 293:** It is essential to mention here the other important players. For example, MSA is never present in the atmosphere without SA as both are important DMS oxidation products.

**Response:** According to the reviewer's pertinent suggestion, the statement about other important players for marine NPF was added in lines 280-283 of the revised manuscript as follows:

"… multi-component nucleation model. For example, both SA and MSA originate from the oxidation of DMS, so their coexistence in the atmosphere may synergistically promote the formation of IA clusters, which is worthy of future studies."
* * *
**Comment 23.**

**Line 307:** Please review the reference list:

■  There are references with missing journal names or abbreviated journal names in the author list. For example, Bates et al. (2020), Elm and Kristensen et al. (2017), Hatakeyama et al. (1982), Takegawa et al. (2020).

■  There are some references that do not have the complete author list. For example, Beck et al. (2020) and He et al (2021).

■  The Seinfeld and Pandis citation is incorrect and refers to Jeffrey Steinfeld's review of the book.

■  Provide a URL for Stewart (2016).

**Response:** Thanks for the reviewer's carefulness review. The above references have been completed and all references have been double-checked.
* * *
**Comment 24.**

**Figure S1:** The caption of this figure could be misleading because the word 'stable' could be

interpreted from the view of having a ratio of collision frequency to total evaporation that is higher than 1 (Fig. 2c). So please replace the word 'stable' with the 'lowest free energy'. Please also include the temperature in the caption.

**Response:** Thanks for the reviewer's valuable suggestion. "…identified stable configurations" has been corrected to "…identified configurations with lowest free energy" in the revised supplement. The temperature has been added in the caption of Figure S1.
* * *
Thanks again for the reviewer's professional and carefulness review. Accordingly, we have tried our best to improve the manuscript.

Sincerely Yours,
Prof. Xiuhui Zhang

**Reference**

Arquero, K. D., Xu, J., Gerber, R. B., and Finlayson-Pitts, B. J.: Particle formation and growth from oxalic acid, methanesulfonic acid, trimethylamine and water: a combined experimental and theoretical study, Phys. Chem. Chem. Phys., 19, 28286–28301, https://doi.org/10.1039/C7CP04468B, 2017.

Beck, L. J., Sarnela, N., Junninen, H., Hoppe, C. J. M., Garmash, O., Bianchi, F., Riva, M., Rose, C., Peräkylä, O., Wimmer, D., Kausiala, O., Jokinen, T., Ahonen, L., Mikkilä, J., Hakala, J., He, X., Kontkanen, J., Wolf, K. K. E., Cappelletti, D., Mazzola, M., Traversi, R., Petroselli, C., Viola, A. P., Vitale, V., Lange, R., Massling, A., Nøjgaard, J. K., Krejci, R., Karlsson, L., Zieger, P., Jang, S., Lee, K., Vakkari, V., Lampilahti, J., Thakur, R. C., Leino, K., Kangasluoma, J., Duplissy, E., Siivola, E., Marbouti, M., Tham, Y. J., Saiz-Lopez, A., Petäjä, T., Ehn, M., Worsnop, D. R., Skov, H., Kulmala, M., Kerminen, V., and Sipilä, M.: Differing Mechanisms of New Particle Formation at Two Arctic Sites, Geophys Res Lett, 48, https://doi.org/10.1029/2020GL091334, 2021.

Bork, N., Elm, J., Olenius, T., and Vehkamäki, H.: Methane sulfonic acid-enhanced formation of molecular clusters of sulfuric acid and dimethyl amine, Atmos. Chem. Phys., 14, 12023–12030, https://doi.org/10.5194/acp-14-12023-2014, 2014.

He, X.-C., Tham, Y. J., Dada, L., Wang, M., Finkenzeller, H., Stolzenburg, D., Iyer, S., Simon, M., Kürten, A., Shen, J., Rörup, B., Rissanen, M., Schobesberger, S., Baalbaki, R., Wang, D. S., Koenig, T. K., Jokinen, T., Sarnela, N., Beck, L. J., Almeida, J., Amanatidis, S., Amorim, A., Ataei, F., Baccarini, A., Bertozzi, B., Bianchi, F., Brilke, S., Caudillo, L., Chen, D., Chiu, R., Chu, B., Dias, A., Ding, A., Dommen, J., Duplissy, J., El Haddad, I., Gonzalez Carracedo, L., Granzin, M., Hansel, A., Heinritzi, M., Hofbauer, V., Junninen, H., Kangasluoma, J., Kemppainen, D., Kim, C., Kong, W., Krechmer, J. E., Kvashin, A., Laitinen, T., Lamkaddam, H., Lee, C. P., Lehtipalo, K., Leiminger, M., Li, Z., Makhmutov, V., Manninen, H. E., Marie, G., Marten, R., Mathot, S., Mauldin, R. L., Mentler, B., Möhler, O., Müller, T., Nie, W., Onnela, A., Petäjä, T., Pfeifer, J., Philippov, M., Ranjithkumar, A., Saiz-Lopez, A., Salma, I., Scholz, W., Schuchmann, S., Schulze, B., Steiner, G., Stozhkov, Y., Tauber, C., Tomé, A., Thakur, R. C., Väisänen, O., Vazquez-Pufleau, M., Wagner, A. C., Wang, Y., Weber, S. K., Winkler, P. M., Wu, Y., Xiao, M., Yan, C., Ye, Q., Ylisirniö, A., Zauner-Wieczorek, M., Zha, Q., Zhou, P., Flagan, R. C., Curtius, J., Baltensperger, U., Kulmala, M., Kerminen, V.-M., Kurtén, T., et al.: Role of iodine oxoacids in atmospheric aerosol nucleation, Science, 371, 589–595, https://doi.org/10.1126/science.abe0298, 2021.

Khanniche, S., Louis, F., Cantrel, L., and Černušák, I.: A theoretical study of the microhydration of iodic acid (HOIO$_2$), Computational and Theoretical Chemistry, 1094, 98–107, https://doi.org/10.1016/j.comptc.2016.09.010, 2016.

McGrath, M. J., Olenius, T., Ortega, I. K., Loukonen, V., Paasonen, P., Kurtén, T., Kulmala, M., and Vehkamäki, H.: Atmospheric Cluster Dynamics Code: a flexible method for solution of the birth-death equations, Atmos. Chem. Phys., 12, 2345–2355, https://doi.org/10.5194/acp-12-2345-2012, 2012.

Rong, H., Liu, J., Zhang, Y., Du, L., Zhang, X., and Li, Z.: Nucleation mechanisms of iodic acid in clean and polluted coastal regions, Chemosphere, 253, 126743, https://doi.org/10.1016/j.chemosphere.2020.126743, 2020.

---

## Author Comment (AC3)

**Responses to Referee #3's comments**

Thanks for the reviewer's professional and helpful comments on our manuscript "**Molecular-level evidence for marine aerosol nucleation of iodic acid and methanesulfonic acid**" (MS No.: acp-2021-595). We have revised the manuscript carefully according to reviewer's comments. The point-to-point responses to the Referee #3's comments are summarized below:

**Major comments:**

**Comment 1:** The authors deployed identical QC methods in this work and their earlier work (Rong et al. 2020; Xiuhui Zhang is the corresponding author for both studies). The authors calculated formation rates for pure IA nucleation in both studies, but a 1000 to 10,000 times difference can be found by comparing Fig. 3 of this study and Fig. 2 in Rong et al. 2020. A further check on Table S12 in Rong et al. 2020 and Table S2 in this study shows substantially different Gibbs free energy values for the same IA clusters at the same temperature. E.g., at 278 K, Rong et al. 2020 gives a $\Delta G$ of -5.92 kcal mol$^{-1}$ while this study gives -9.51 kcal mol$^{-1}$. The difference seems small but as it goes in the exponential part of the evaporation rate equation, the resulted evaporation rates can be significantly different. My simplified calculation suggests that a -9.51 kcal mol-1 value easily results in a 500 times lower evaporation rate of iodic acid dimer compared to a value of -5.92.

A further investigation on this matter comparing these two papers suggest at least two major differences. Rong et al. 2020 uses a double zeta basis set (aug-cc-pVDZ-PP) while this study uses a triple-zeta basis set (aug-cc-pVTZ-PP). However, I highly doubt that this is the primary reason for such a significant difference. I further tried to compare the geometries of the IA dimer in these two papers. As the authors provided wrong coordinates for IA dimer in Supp-Section 6 in Rong et al. 2020, I have to infer from their Fig. S3 (cited here as Figure R1). The IA dimer is connected by two halogen bonds. However, in this study, the IA dimer is connected by two hydrogen bonds from my reproduced results based on coordinates in Table S7 of this manuscript. Additionally, the geometries of larger IA clusters in this study are also significantly different from Rong et al. 2020. The authors should visualize the IA clusters and any other clusters which are not yet visualized besides IA-MSA clusters in Fig S1.

Because of these significant differences in these two papers, the authors are urged to do at least as follows: 1) they should calculate all the commonly used clusters (at least all of the IA clusters) in Rong et al. 2020 and this study by both the double zeta and triple zeta basis sets for two sets of geometries provided in both papers (so 2x2 matrix for every cluster). 2) the authors should discuss the results coming from item 1) in the Main Text and give reasons for their updated

results and potential errors associated with the Rong et al. 2020 or this study. 3) The authors are encouraged also to discuss in the main text why the geometries provided in their current study should represent global minima and why the geometries are significantly different in Rong et al. 2020 and this study.

[Figure]

Figure R1. Screenshot of Figure S3 in Rong et al. 2020.

**Response:** Thanks for the reviewer's professional and valuable comments. The responses to each of the reviewer's comments are presented below.

**Item 1) from the reviewer**: "they should calculate all the commonly used clusters (at least all of the IA clusters) in Rong et al. 2020 and this study by both the double zeta and triple zeta basis sets for two sets of geometries provided in both papers (so 2x2 matrix for every cluster)."

**Response:** As suggested by the reviewer, all of the studied IA clusters in Rong et al. 2020 and this study have been calculated at both double zeta and triple zeta basis sets. The resulting $\Delta G$s at 278K of IA clusters are collected in the following Table A1, which has also been added in the revised supporting file (Table S2).

Table A1. The Gibbs formation free energies $\Delta G_{278K}$ (kcal mol$^{-1}$) of the studied IA clusters in Rong et al. 2020 and this study calculated at the RI-CC2/aug-cc-pVTZ(-PP)//$\omega$B97X-D/6-311++G(3df,3pd) + aug-cc-pVDZ-PP (for I) (DZ) and RI-CC2/aug-cc-pV(T+d)Z(-PP)//$\omega$B97X-D/6-311++G(3df,3pd) + aug-cc-pVTZ-PP (for I) (TZ), respectively.

| Cluster | $\Delta G_{278K}$-Rong (DZ) | $\Delta G_{278K}$-Rong (TZ) | $\Delta G_{278K}$-this study (DZ) | $\Delta G_{278K}$-this study (TZ) |
|---------|------|------|------|------|
| $(IA)_2$ | -5.92 | -8.07 | -7.96 | -9.51 |
| $(IA)_3$ | -15.73 | -18.47 | -16.05 | -18.79 |
| $(IA)_4$ | -34.41 | -38.48 | -34.41 | -38.48 |
| $(IA)_5$ | -52.37 | -56.28 | -52.37 | -56.28 |
| $(IA)_6$ | -70.67 | -74.41 | -70.67 | -74.41 |

It is worth noting that the $(IA)_2$ and $(IA)_3$ clusters in Rong et al. 2020 and this study are different. Specifically, the $(IA)_2$ in Rong is halogen-bonded, while in this study it is hydrogen-bonded. And this issue will be further discussed in the response to item 2) below.

**Item 2) from the reviewer**: "the authors should discuss the results coming from item 1) in the Main Text and give reasons for their updated results and potential errors associated with the Rong et al. 2020 or this study."

**Response:** Thanks for the reviewer's valuable suggestion. The following are **A)** the causes of the differences in values, **B)** the reasons for updating the data , and **C)** the potential errors in the Rong et al. 2020 or this study.

A)  In the Rong et al. 2020 and this study, the levels of theory employed in cluster structure optimization and frequency calculations are different. The larger triple-$\zeta$ basis set (aug-cc-pVTZ-PP (for iodine)) was used in this study compared to the double-$\zeta$ basis set (aug-cc-pVDZ-PP (for iodine)) of Rong et al., (2020). As shown in the above Table A1 (second and third columns), the different basis sets lead to differences in the calculated $\Delta G$s of IA clusters $((IA)_{2-6})$ in the range of $2.15 \sim 4.06$ kcal mol$^{-1}$. This is one reason for the difference in the calculated $\Delta G$s values.

In addition, as mentioned above and shown in Fig. A1, the geometries of IA clusters (IA dimer and IA trimer) employed in the Rong et al. 2020 and the present manuscript are different. This is another reason. It further leads to a difference of $0.32 \sim 1.44$ kcal mol$^{-1}$ of $\Delta G$s values , at the same level of theory (third and fifth columns in Table A1).

[Figure]

Rong et al. (2020)                    The present manuscript

$(IA)_2$                              $(IA)_2$

$(IA)_3$                              $(IA)_3$

Figure A1. The different IA cluster structures employed in Rong et al. (2020) and the present manuscript (including IA dimer and trimer).

B) The following are the reasons for the choice of the double-ζ basis set in the Rong et al. 2020, and the subsequent update of the results in the present paper:

Due to large number of electrons in iodine atom, the QC calculations of clusters involving IA are expensive. Considering the variety of clusters calculated in the Rong et al. 2020, coupled with the limited computational resources available at that time, the double-ζ basis set was chosen as a compromise between computational accuracy (Benchmark, Table S1-S6 in Rong et al. 2020) and resource consumption to present the reasonable trends and corresponding mechanisms of sulfuric acid and $NH_3$ promoting IA cluster formation. After updating the computational resources, in the present manuscript, the larger triple-ζ basis set was herein used to reduce errors in the subsequent dynamic simulations because higher level of theory usually implies a better calculation accuracy.

As suggested by the reviewer, we have declared the reason for updating the results in the lines 66-69 of the revised manuscript.

C) The potential errors in the Rong et al. 2020 or this study are as follows:

Thanks for the reviewer's professional comments. After careful comparison (Table A1) we found that with the same structure, the $\Delta G$ calculated in Rong et al. 2020 at the double-ζ basis set (aug-cc-pVDZ-PP for iodine) is higher than those under triple-ζ basis set (aug-cc-pVTZ-PP for iodine), which will lead to a lower cluster formation rate.

**Item 3) from the reviewer**: "The authors are encouraged also to discuss in the main text why the geometries provided in their current study should represent global minima and why the geometries are significantly different in Rong et al. 2020 and this study."

**Response:** Thanks for the reviewer's helpful suggestion. The reasons for the selection of the different IA cluster conformations (IA dimer and trimer) in the Rong et al. 2020 and this work are summarized below:

The sampling process of cluster isomers is computationally demanding and subject to uncertainties (Elm et al., 2020). The employed IA dimer and trimer in this manuscript are not the lowest-energy isomers due to our incomplete consideration. Thanks for the reviewer's careful review. The IA dimer and trimer with lowest energy obtained after recalculation are consistent with those of Rong et al. 2020. In addition, we have also checked the other employed IA clusters and confirmed that these clusters were selected with the lowest energies.

Thanks again for the reviewer's professional comment. We have corrected the structure of IA dimer and trimer to be consistent with that of Rong et al. 2020. Accordingly, we have also

recalculated all relevant data and updated graphs and tables in the revised manuscript and revised supporting file.

**Item 4) from the reviewer**: "The authors should visualize the IA clusters and any other clusters which are not yet visualized besides IA-MSA clusters in Fig. S1."

As suggested by the reviewer, all the clusters studied in the present manuscript, containing IA clusters as well as other previously unpresented clusters, have been visualized in the revised Fig. S1 of the supplement. In addition, by a comparison of with the original data, we find that the mentioned Cartesian coordinates of IA dimer and trimer provided in SI of Rong et al. (2020) lost the corresponding negative sign due to typos. And the corrected coordinates are added in the Appendix in this response. Further, we are contacting the corresponding publisher to correct this issue in these days.
* * *
**Comment 2:** Another fundamental problem is that neither this study nor Rong et al. 2020 seem to even remotely repeat CLOUD measurements on pure iodic acid nucleation (He et al. 2021). For example, this study calculates 6 orders of magnitude lower cluster formation rates compared to He et al. 2021 and the difference goes to 9 orders when comparing Rong et al. 2020 with He et al. 2021. While it is understandable that the large number of electrons in iodine atom cause substantial challenges in QC calculations, such a substantial difference must be explained, as it could potentially suggest that either there are some fundamental errors in the QC + ACDC calculations related to coding, methods and basis sets employed or because the authors are calculating based on wrong assumptions.

**Response:** Thanks for the reviewer's constructive suggestions. We realized that there is a significant discrepancy between the simulated rates in the present manuscript and the CLOUD measurements (He et al. 2021). The reasons for this phenomenon are as follows (A and B):

A) The nucleation components involved in the ACDC simulation (only IA) of the present manuscript and CLOUD experiment (He et al., 2021) are different. If we understand correctly, He et al., (2021) presented a high dependence of the nucleation rate on the concentration of IA (Fig. 1, He et al., 2021), but in fact, not only IA is involved in nucleation, but also other iodine components, such as $I_2O_4$, $I_2O_5$, and $HIO_2$, play a non-negligible role in forming IA particles (Fig. 2 and Fig. 3, He et al., 2021). Therefore, using the simulated rate containing only IA to compare the experimental nucleation rates of multiple iodine components such as $I_2O_4$, $I_2O_5$, and $HIO_2$, will inevitably lead to large differences.

B) In addition to the differences in the components mentioned in item 1), as the reviewer

mentioned, there are uncertainties in both experiments and simulations. The superposition of errors may further amplify the difference between experiments and simulations. In addition, there are also differences in the way nucleation rates are calculated between the experiments and simulations. All of these would lead to the fact that we cannot simply assess the cause of the discrepancy and attribute it to one point.

Therefore, there is uncertainty in comparing the specific results of experiments and simulations in this case. In fact, theoretical calculations prefer to show that the IA-MSA system have higher cluster formation rates than the pure-IA system, indicating the enhancement of MSA on IA cluster formation, which is particularly evident in marine regions with rich MSA and sparse IA.
* * *
**Comment 3:** I'm not convinced that there are sufficient IA-MSA clusters calculated in this study which would allow the authors to consider larger clusters than IA4MSA2 and IA6 as nucleated clusters. The largest ratios of growth to evaporation in Fig. 2c are 2 (IA4MSA2) and 7 (IA6) which barely provide growth potential for these clusters. How can these clusters be considered stable enough? This even more true when looking at the Fig. S2C in which the largest value is 0.2. This is an essential assumption for this study and all other QC + ACDC studies and many of the numbers in this study will make no sense if this is not varied. The author should extend their calculations until finding stable clusters.

Additionally, the authors mistakenly conclude in lines 171-172 that "condensation sink" has a minor impact on the cluster formation rate. First, likely the authors are not talking about condensation sink but coagulation loss or a combination of other losses. This should be clarified, and they should describe clearly that whether they applied the "CS" uniformly for all clusters. Typical in ACDC models one would set monomer species as constants and therefore condensation sink does not affect the monomer condensation. If their calculations correctly find stable clusters with low enough evaporation rates (and thus high growth potential from condensation), they will likely find the "CS" as an important factor influencing their calculated cluster formation rates. It's possible that they find such odd results because they have not found the clusters with low enough evaporation rates as mentioned above. This is evident from the fact that if stable clusters are found, condensational growth is likely dominating the growth and its value is comparable to their "CS" values of 1e-4 to 2.6e-3 s$^{-1}$ when acid is ranging from 1e6 to 1e8 cm$^{-3}$. The lowest evaporations rates given in this study (Fig. 2B) are not far from the "CS".

**Response:** Thanks for the reviewer's professional comment. The specific response to each item

of the reviewer are as follows:

**Item 1) from the reviewer**: "The largest ratios of growth to evaporation in Fig. 2c are 2 (IA4MSA2) and 7 (IA6) which barely provide growth potential for these clusters. How can these clusters be considered stable enough?"

**Response:** In ACDC simulations (McGrath et al., 2012), relatively stable clusters are those in which collisions with molecules can be assumed to dominate over cluster evaporation (Oona and Tinja, 2020). Specifically, it is to calculate the ratio of the rate at which the studied cluster collides with the IA or MSA monomer to its total evaporation rate. The following is an example of the IA4MSA2 cluster.

$$\frac{\beta_I C_I}{\sum \gamma} = \frac{\beta_I C_I \cdot C_{IA4MSA2}}{\sum \gamma_{IA4MSA2} \cdot C_{IA4MSA2}} = 2 > 1$$

where $\beta_I$ is the rate coefficient of cluster collision with IA monomer, $C_I$ is the concentration of IA monomer, and $\sum \gamma$ is the total evaporation rate coefficient of the studied cluster.

If $\beta_I C_I / \sum \gamma > 1$, the corresponding cluster would be considered to be relatively stable against evaporation, and has the "growth potential". Once these stable clusters on the boundary form and grow further out of the simulated system, these formed clusters out of system are unlikely evaporate back into the system (Oona and Tinja, 2020; McGrath et al., 2012). Thus, as suggested by the reviewer, the controversial statement "stable enough" has been changed to "relatively stable" in the revised manuscript. In addition, the above explanation has been added in lines 162 – 167 (main text) and Section S1 (supplement) for the clarity of the reader.

**Item 2) from the reviewer**: This even more true when looking at the Fig. S2C in which the largest value is 0.2. This is an essential assumption for this study and all other QC + ACDC studies and many of the numbers in this study will make no sense if this is not varied.

**Response:** This is a very helpful point – thank you for bringing it up. In fact, the present ratios of growth to evaporation ($\beta_I C_I / \sum \gamma$) in Fig. 2C, Fig. S2C and Fig. S3C are the lowest values, since the chosen $C_I$ at this point is the lowest concentration of IA monomer ($1.0 \times 10^6$ molecules cm$^{-3}$). The $\beta_I C_I / \sum \gamma$ of the mentioned cluster is in the range of 0.2 - 20 under the studied range of IA concentration ($10^6$ - $10^8$ molecules cm$^{-3}$). In this case, the clusters (value of 0.2 at [IA] = $1.0 \times 10^6$ molecules cm$^{-3}$) can grow out of the system at most of studied [IA] ($5 \times 10^6 \sim 10^8$ molecules cm$^{-3}$.

Thanks to the professional advice of the reviewer. In order to let the reader know that the presented $\beta_I C_I / \sum \gamma$ is the minimum value, we have added notes in lines 163 – 165 (main text) and Section S1 (supplement) to remind the reader.

**Item 3) from the reviewer**: "First, likely the authors are not talking about condensation sink but coagulation loss or a combination of other losses. This should be clarified, and they should describe clearly that whether they applied the "CS" uniformly for all clusters."

**Response:** As pertinently suggested by the reviewer, the statement about "CS" uniformly for all clusters has been added in line 184 of the revised manuscript. This treatment is also commonly used in other theoretical simulation studies (Bork et al., 2014; Shen et al., 2019; Xu et al., 2020).

**Item 4) from the reviewer**: "Additionally, the authors mistakenly conclude in lines 171-172 that "condensation sink" has a minor impact on the cluster formation rate… .

**Response:** In this study, before getting the corresponding conclusions ("the different CS $(1.0\times10^{-4} \sim 2.6\times10^{-3}$ s$^{-1})$ have a minor impact on the cluster formation rate"), we have tested the effect of different CS values on cluster formation rate $J$ (cm$^{-3}$ s$^{-1}$) in Fig. S4 of supporting file. For the reviewer's convenience, we have copied and presented it as follows.

[Figure]

**Figure S4.** The simulated cluster formation rate $J$ (cm$^{-3}$ s$^{-1}$) of the IA-MSA system at different condensention sink (CS) coefficients (CS = $1.0\times10^{-4} \sim 2.6\times10^{-3}$ s$^{-1}$), $T = 278$ K, [IA] = $10^{6} \sim 10^{8}$ molecules cm$^{-3}$, and [MSA] = $10^{6}$ (blue lines), $10^{8}$ (orange lines) molecules cm$^{-3}$.

As the reviewer expertly suggested, as shown in Fig. 4, different CS values indeed have an impact on $J$. In the case of higher [IA] and [MSA], the $J$ is relatively high and the effect of CS is relatively little. However, the effects of different CS values are more pronounced in the case of low $J$ because the order of magnitude of $J$ at this moment is comparable to or lower than the employed CS values. To ensure the rigor of the statement, we have changed the "the different CS have a minor impact on the cluster formation rate" to the "the different CS have an impact on the simulated $J$, especially in the case of low $J$ (Fig. S4), but less on presenting the promotion of MSA on IA cluster formation and the main conclusions of this study." in lines 182-184 of

the revised manuscript. Thanks again for the reviewer's kind reminder.
* * *
**Specific comments:**

**Comment 4.** The authors mention three important studies (Sipilä et al. 2016, Baccarini et al. 2020 and Beck et al. 2020) as the backbone of this study. However, none of these studies support their calculations. Sipilä et al. 2016 and Baccarini et al. 2020 indicate that IA NPF dominate the NPF events they observed at Mace Head and the central Arctic, respectively, while Beck et al. 2020 suggests SA-NH3 is the dominating nucleation mechanism in Ny Alesund by comparing SA with nucleation rates. They do, however, suggest MSA contributes to particle growth. These studies go against what the authors suggest.

Lines 273-274: this is not enough to respond my comments in the initial screening. Agreeing with the importance of other molecules goes against their title, and thus the main theme of this manuscript. If I understand correctly from the literature, other iodine species can be formed at the same time as iodic acid. Similarly, both MSA and SA are formed from DMS in marine environments. These species are very likely to co-exist in marine environments at different levels. Picking up two species from the list and claim it to be a marine aerosol nucleation mechanism is not acceptable unless it is either supported by their calculations or by field observations. However, the mentioned three studies clearly disagree with the IA-MSA nucleation mechanisms. Additionally, the cluster formation rates derived in this study are too low to explain field observations (details below).

**Response:** Thanks for your pertinent feedback and suggestions. The reviewer's comments and corresponding responses are summarized below.

**Item 1) from the reviewer**: "The authors mention three important studies (Sipilä et al. 2016, Baccarini et al. 2020 and Beck et al. 2020) as the backbone of this study. However, none of these studies support their calculations."

**Response:** We agree with you that the three mentioned references (Sipilä et al., 2016; Baccarini et al., 2020; Beck et al., 2021) do not support the importance of the IA-MSA mechanism. However, the findings of these studies were based on observations obtained in specific areas and do not inherently conflict with the results presented in this manuscript. The reasons are as follows (A, B, and C):

    A) For Mace Head (Sipilä et al., 2016), the simulations were only compared with the observations at this site in Fig. 5b of original manuscript. The simulation results show that the IA-MSA mechanism plays a much smaller role of 1% in Mace Head where IA

concentrations can be as high as $10^8$ molecules cm$^{-3}$. In the lines 238 - 239 of preprint manuscript, we have also emphasized that for regions with high IA concentrations like Mace Head, the contribution of the IA-MSA mechanism is minor and the NPF process remains dominated by IA. This conclusion does not seem to conflict with Sipilä et al. (2016).

B) During the Arctic Ocean 2018 expedition (Baccarini et al., 2020), the simulation results based on the field conditions ($T$ = 268 K, CS = 1×10$^{-4}$ s$^{-1}$, [IA] = $10^5$~$10^6$ and [MSA] = $10^5$ molecules cm$^{-3}$) in the present manuscript shows a contribution of 1% - 16% to cluster formation from the IA-MSA mechanism. Most of the contribution is attributed to nucleation of IA (declared in lines 250 - 254 of the preprint manuscript). This seems not to be in conflict with the conclusion that the NPF observed by Baccarini et al. (2020) is mainly driven by IA.

C) As to Ny-Ålesund (Beck et al., 2020), the present results in Fig. 6 suggest that the simulated rate of IA-MSA mechanism only partially matches a small fraction of the observed rates, and most of the unmatched rate might come from the contribution of SA-NH$_3$ ion-induced nucleation (declared in lines 270 - 272 of the preprint manuscript). This result seems also not conflicting.

In addition to the mentioned specific regions with higher [IA] and relatively lower [MSA], considering the vastness of the ocean, the simulations presented in this manuscript suggest that the IA-MSA mechanism could potentially play an important role in regions with sparse IA and rich MSA.

Thanks again for the reviewer's careful review and valuable comment. To add, the original intention of citing the work of Sipilä et al. (2016) and Baccarini et al. (2020) in the background was only to show that IA is a key NPF driver in coastal and polar regions, not to support the importance of IA-MSA mechanisms in these regions. And in the section of rate analysis, the citation of Sipilä et al. (2016) aims to support the reasonableness of the IA atmospheric concentration range employed in the simulations, rather than compare with the corresponding field observation.

**Item 2) from the reviewer**: "Similarly, both MSA and SA are formed from DMS in marine environments. These species are very likely to co-exist in marine environments at different levels. Picking up two species from the list and claim it to be a marine aerosol nucleation mechanism is not acceptable unless it is either supported by their calculations or by field observations. However, the mentioned three studies clearly disagree with the IA-MSA

nucleation mechanisms. Additionally, the cluster formation rates derived in this study are too low to explain field observations (details below)".

**Response:** Indeed, as the reviewer expertly suggested, there are many components in the real atmosphere besides IA and MSA, such as SA or other iodine components that may participate in the nucleation process together. However, this study focuses more on the impact of MSA, a representative oceanic sulfur-containing acid. And other components will be taken into account in future research.
* * *
**Comment 5.** Fig 6: this figure is misleading. The authors overlooked the conclusions from Beck et al. 2020 which suggested SA-NH3 to be the nucleating mechanism in Ny-Alesund, supported by measurement data. The authors adopted the cluster formation rates from Beck et al. 2020 and presented it as a rectangle in the plot which falsely overlaps with their data. If iodic acid is nucleating with MSA in Ny Alesund, one would expect that the cluster formation rates go up with increasing iodic acid. If the data are correctly presented, the authors will find that their data are very far away from the field observation they presented. All the discussions following this figure are therefore problematic. If I have to assume that the authors do calculate the IA-MSA cluster formation rates correctly (which I doubt as presented below), the results suggest that the IA-MSA mechanism does not play a role in Ny Alesund thus agreeing with Beck et al. 2020.

**Response:** Thanks for the helpful comments. As professionally suggested by the reviewer, we realized that the cluster formation rate of the IA-MSA mechanism simulated in Fig. 6 is at variance with the reported rate (Beck et al. 2020). Thus, the controversial Fig. 6 has been removed from the revised manuscript.

In addition, we have rewritten the corresponding results section (lines 240 – 273 in the revised manuscript) to focus more on the simulation results rather than any comparison with the field observation.
* * *
**Comment 6.** 5B: the authors cited iodic acid data from another study and assumed an MSA value of 1e7 cm-3. While I agree with the authors that there are not enough MSA measurements around the world, the authors should not overclaim their quantitative results because of their huge uncertainties. Additionally, many other species are not considered in the current mechanism which further deepens their discrepancy. The numbers in this plot are repeated in the Discussions and are presented as the key results in the Results part. This will be very misleading for future readers and should be all removed since there is no way for this study to

get any remotely correct estimations.

**Response:** As professionally suggested by the reviewer, the simulations in Fig. 5B would indeed overestimate the impact of the IA-MSA mechanism on regions with MSA concentrations below $10^7$ molecules cm$^{-3}$. We agree with you and have therefore removed Fig. 5B from the revised manuscript. Moreover, to present the results that the contribution of the IA-MSA mechanism is highly dependent on the MSA and IA concentration, the contribution of the different growth pathway varying with [MSA] ($10^6$ –$10^7$ molecules cm$^{-3}$) and [IA] ($10^6$ –$10^8$ molecules cm$^{-3}$) has been presented in the redrawn Fig. 5b, as follows:

[Figure]

**Figure 5. (a)** Main cluster growth pathway of IA-MSA nucleating system at $T$ = 278 K, CS = $2.0 \times 10^{-3}$ s$^{-1}$, [IA] = $10^7$ and [MSA] = $5 \times 10^6$ molecules cm$^{-3}$. The black and orange arrows refer to the pathways of colliding with IA and MSA, respectively, where the dashed arrows indicate the evaporation of MSA. **(b)** Branch ratio of IA-MSA (orange pie) and pure-IA (purple pie) growth pathway under varying [MSA] ($10^6$ – $10^7$ molecules cm$^{-3}$) and [IA] ($10^6$ – $10^8$ molecules cm$^{-3}$).

In addition, given that the results at [MSA] = $10^7$ molecules cm$^{-3}$ would overestimate the impact of the IA-MSA mechanism on some sites, the relevant statement has been removed from the revised manuscript.
* * *
**Minor comments:**

**Line13-16:** repeating message.

**Response:** Thanks for the reviewer's carefulness reading. The repeating message has been removed and integrated in lines 13 – 15 of revised manuscript as follows: "Moreover, MSA can promote IA clusters formation, particularly in cold marine regions with sparse IA and rich MSA. For the IA-MSA nucleation mechanism, in addition to self-nucleation of IA, the IA-MSA-involved clusters can also directly participate in the nucleation process."

**Line24:** ground based open ocean nucleation was not frequently measured but relatively rarely

**Response:** Thanks for the reviewer's suggestion, the "frequent" has been removed in the revised manuscript.
* * *
**Line29:** Marine NPF, particularly in remote areas, is more …

**Response:** Accordingly, the corresponding sentence has been changed to "Marine NPF, particularly in remote marine areas, is more affected by biological emissions compared to …" in line 26 of the revised manuscript.
* * *
**Line40:** The coastal NPF and low tide correlation was already established back to the early 2000s, e.g., O'Dowd, 2002.

**Response:** According to the reviewer's professional suggestion, the corresponding reference for O'Dowd, 2002 has been added in line 38 of the revised manuscript.
* * *
**Line45:** I do not find particle composition measurements of MSA and IA in Beck et al. 2020?

**Response:** In fact, in the study of Beck et al. 2020, the original statement is "At Ny-Ålesund, MSA, IA and nitrate/nitric acid are detected in some of the smallest clusters…" in the caption of Fig. 3 (Beck et al., 2020). Herein, according to the study (Beck et al., 2020), the "MSA and IA found in particles" has been corrected to "MSA and IA were detected in the smallest clusters" in line 45 of the revised manuscript.
* * *
**Line47:** mechanism should either be singular to plural.

**Response:** Thanks for the reviewer's suggestion. For consistency, the "mechanisms" has been changed to "mechanism".
* * *
**Line53**: in marine environments / in marine regions

**Response:** According to the reviewer's suggestion, the "at marine regions" has been changed to "in marine environments".
* * *
**Line92:** "t" to "the"

**Response:** "t" has been corrected to "the".
* * *
**Line143:** boundary layer to troposphere? Tropopause? Stratosphere? Clarify.

Fig. 2 and throughout this manuscript. Either use molecules cm-3 or pptv throughout this manuscript. Don't mix these two units.

**Response:** Thanks for the reviewer's professional comment. According to the previous study (Williamson et al., 2019), it is "boundary layer to free troposphere". As suggested by the reviewer, the unit (pptv) used in Fig. 2 has been changed to "molecules cm$^{-3}$" throughout this manuscript to ensure uniformity of units.
* * *
**Line 152**: explain beta and C. What are the subscriptions for beta and C in Fig 2C?

Fig.4: write the exact conditions (fixed values) for the three sets of simulations explicitly in the caption.

**Response:** Thanks for the reviewer's helpful suggestion. In Fig. 2C, the $\beta$ and $C$ represent the collision rate coefficient and monomer concentration, respectively. The subscription 'I' for $\beta_I C_I$ represents IA. Specially, $\beta_I$ is the rate coefficient of cluster collision with IA monomer, and $C_I$ is the concentration of IA monomer. These explanations have been added in the caption of Fig. 2 (lines 173 – 175 of the revised manuscript).

   In addition, according to the reviewer's suggestion, the exact conditions (fixed values) for the three sets of simulations in Fig. 4 has been added in the corresponding caption (lines 204 - 205).
* * *
**Line 204**: it is not clear to me why reduced collision rates result in reduced $R$ since all collision rates are reduced not just collisions between MSA and IA.

**Response:** Thanks! This is a valuable suggestion from the reader's perspective. I am sorry for our unclear presentation. Indeed, as the reviewer expertly analyzed, the decrease in temperature would reduce the intermolecular collision between IA and MSA, as well as IA and IA. The cause of $R$ decay needs to be analyzed in terms of its definition as follows:

$$R = \frac{J_{\text{IA-MSA}}}{J_{\text{pure-IA}}} = \frac{J(\text{IA-MSA cluster + pure-IA cluster})}{J(\text{pure-IA cluster})}$$

The numerator term $J_{\text{IA-MSA}}$ is affected by the formation of both pure-IA clusters and IA-MSA clusters. While the denominator term $J_{\text{pure-IA}}$ is only affected by the formation of pure-IA clusters. Therefore, when the overall collision rate between IA and IA, as well as IA and MSA, is reduced due to the decrease in temperature, the numerator would be affected more than the denominator, which in turn leads to a reduced $R$. For the avoidance of confusion, the corresponding explanation has been added in lines 218 - 221 of the revised manuscript.
* * *
**Line 221**: these findings. I don't find MSA and IA detected in the particle phase in Beck et al. 2020. Could the authors specify where I can find the information there?

**Response:** Thanks for the reviewer's helpful suggestion. The restatement in the present manuscript is not accurate and should be "smallest clusters" rather than "particles". The corresponding statement in the work of Beck et al. 2020 is in the caption of Fig. 3 of "At Ny-Ålesund, MSA, IA and nitrate/nitric acid are detected in some of the smallest clusters…".
* * *
**Line 233**: the MSA concentrations are likely not always this high in these sites. Could the authors give a bit more reasoning in their choice of 1e7 cm-3?

**Response:** Thanks for your valuable comment. The reason is that according to the global distribution of MSA simulated by GEOS-Chem model (Chen et al., 2018), MSA concentrations can reach ppt level in most marine areas (Fig. 2(d) in Chen et al. 2018) including our studied regions. Hence, the MSA concentration employed in the simulations was chosen at the order of $10^7\,\text{cm}^{-3}$.

   As professionally suggested by the reviewer, the simulations in Fig. 5b ([MSA] = $10^7$ molecules cm$^{-3}$) would overestimate the impact of the IA-MSA mechanism on regions with concentrations below $10^7$ molecules cm$^{-3}$. Considering the fact that the MSA concentration of $10^7$ molecules cm$^{-3}$ is not applicable for some marine regions, Fig. 5b has been removed from the main text.
* * *
   Thanks again for the reviewer's professional and carefulness review. Accordingly, we have tried our best to improve the manuscript.

Sincerely Yours,

Prof. Xiuhui Zhang

**Reference**

Baccarini, A., Karlsson, L., Dommen, J., Duplessis, P., Vüllers, J., Brooks, I. M., Saiz-Lopez, A., Salter, M., Tjernström, M., Baltensperger, U., Zieger, P., and Schmale, J.: Frequent new particle formation over the high Arctic pack ice by enhanced iodine emissions, Nat. Commun., 11, 4924, https://doi.org/10.1038/s41467-020-18551-0, 2020.

Beck, L. J., Sarnela, N., Junninen, H., Hoppe, C. J. M., Garmash, O., Bianchi, F., Riva, M., Rose, C., Peräkylä, O., Wimmer, D., Kausiala, O., Jokinen, T., Ahonen, L., Mikkilä, J., Hakala, J., He, X., Kontkanen, J., Wolf, K. K. E., Cappelletti, D., Mazzola, M., Traversi, R., Petroselli, C., Viola, A. P., Vitale, V., Lange, R., Massling, A., Nøjgaard, J. K., Krejci, R., Karlsson, L., Zieger, P., Jang, S., Lee, K., Vakkari, V., Lampilahti, J., Thakur, R. C., Leino, K., Kangasluoma, J., Duplissy, E., Siivola, E., Marbouti, M., Tham, Y. J., Saiz-Lopez, A., Petäjä, T., Ehn, M., Worsnop, D. R., Skov, H., Kulmala, M., Kerminen, V., and Sipilä, M.: Differing Mechanisms of New Particle Formation at Two Arctic Sites, Geophys. Res. Lett, 48, https://doi.org/10.1029/2020GL091334, 2021.

Bork, N., Elm, J., Olenius, T., and Vehkamäki, H.: Methane sulfonic acid-enhanced formation of molecular clusters of sulfuric acid and dimethyl amine, Atmos. Chem. Phys., 14, 12023–12030, https://doi.org/10.5194/acp-14-12023-2014, 2014.

Chen, Q., Sherwen, T., Evans, M., and Alexander, B.: DMS oxidation and sulfur aerosol formation in the marine troposphere: a focus on reactive halogen and multiphase chemistry, Atmos. Chem. Phys., 18, 13617–13637, https://doi.org/10.5194/acp-18-13617-2018, 2018.

Elm, J., Kubečka, J., Besel, V., Jääskeläinen, M. J., Halonen, R., Kurtén, T., and Vehkamäki, H.: Modeling the formation and growth of atmospheric molecular clusters: A review, Journal of Aerosol Science, 149, 105621, https://doi.org/10.1016/j.jaerosci.2020.105621, 2020.

He, X.-C., Tham, Y. J., Dada, L., Wang, M., Finkenzeller, H., Stolzenburg, D., Iyer, S., Simon, M., Kürten, A., Shen, J., Rörup, B., Rissanen, M., Schobesberger, S., Baalbaki, R., Wang, D. S., Koenig, T. K., Jokinen, T., Sarnela, N., Beck, L. J., Almeida, J., Amanatidis, S., Amorim, A., Ataei, F., Baccarini, A., Bertozzi, B., Bianchi, F., Brilke, S., Caudillo, L., Chen, D., Chiu, R., Chu, B., Dias, A., Ding, A., Dommen, J., Duplissy, J., El Haddad, I., Gonzalez Carracedo, L., Granzin, M., Hansel, A., Heinritzi, M., Hofbauer, V., Junninen, H., Kangasluoma, J., Kemppainen, D., Kim, C., Kong, W., Krechmer, J. E., Kvashin, A., Laitinen, T., Lamkaddam, H., Lee, C. P., Lehtipalo, K., Leiminger, M., Li, Z., Makhmutov, V., Manninen, H. E., Marie, G., Marten, R., Mathot, S., Mauldin, R. L., Mentler, B., Möhler, O., Müller, T., Nie, W., Onnela, A., Petäjä, T., Pfeifer, J., Philippov, M., Ranjithkumar, A., Saiz-Lopez, A., Salma, I., Scholz, W., Schuchmann, S., Schulze, B., Steiner, G., Stozhkov, Y., Tauber, C., Tomé, A., Thakur, R. C., Väisänen, O., Vazquez-Pufleau, M., Wagner, A. C., Wang, Y., Weber, S. K., Winkler, P. M., Wu, Y., Xiao, M., Yan, C., Ye, Q., Ylisirniö, A., Zauner-Wieczorek, M., Zha, Q., Zhou, P., Flagan, R. C., Curtius, J., Baltensperger, U., Kulmala, M., Kerminen, V.-M., Kurtén, T., et al.: Role of iodine oxoacids in atmospheric aerosol nucleation, Science, 371, 589–595, https://doi.org/10.1126/science.abe0298, 2021.

McGrath, M. J., Olenius, T., Ortega, I. K., Loukonen, V., Paasonen, P., Kurtén, T., Kulmala, M., and Vehkamäki, H.: Atmospheric Cluster Dynamics Code: a flexible method for solution of the birth-death equations, Atmos. Chem. Phys., 12, 2345–2355, https://doi.org/10.5194/acp-12-2345-2012, 2012.

Oona, K.-M. and Tinja, O.: Atmospheric Cluster Dynamics Code Technical manual, https://github.com/tolenius/ACDC/blob/main/ACDC_Manual_2020_11_25.pdf, 2020.

Rong, H., Liu, J., Zhang, Y., Du, L., Zhang, X., and Li, Z.: Nucleation mechanisms of iodic acid in clean and polluted coastal regions, Chemosphere, 253, 126743, https://doi.org/10.1016/j.chemosphere.2020.126743, 2020.

Shen, J., Xie, H.-B., Elm, J., Ma, F., Chen, J., and Vehkamäki, H.: Methanesulfonic Acid-driven New Particle Formation Enhanced by Monoethanolamine: A Computational Study, Environ. Sci. Technol., 53, 14387–14397, https://doi.org/10.1021/acs.est.9b05306, 2019.

Sipilä, M., Sarnela, N., Jokinen, T., Henschel, H., Junninen, H., Kontkanen, J., Richters, S., Kangasluoma, J., Franchin, A., Peräkylä, O., Rissanen, M. P., Ehn, M., Vehkamäki, H., Kurten, T., Berndt, T., Petäjä, T., Worsnop, D., Ceburnis, D., Kerminen, V.-M., Kulmala, M., and O'Dowd, C.: Molecular-scale evidence of aerosol particle formation via sequential addition of HIO3, Nature, 537, 532–534, https://doi.org/10.1038/nature19314, 2016.

Williamson, C. J., Kupc, A., Axisa, D., Bilsback, K. R., Bui, T., Campuzano-Jost, P., Dollner, M., Froyd, K. D., Hodshire, A. L., Jimenez, J. L., Kodros, J. K., Luo, G., Murphy, D. M., Nault, B. A., Ray, E. A., Weinzierl, B., Wilson, J. C., Yu, F., Yu, P., Pierce, J. R., and Brock, C. A.: A large source of cloud condensation nuclei from new particle formation in the tropics, Nature, 574, 399–403, https://doi.org/10.1038/s41586-019-1638-9, 2019.

Xu, C.-X., Jiang, S., Liu, Y.-R., Feng, Y.-J., Wang, Z.-H., Huang, T., Zhao, Y., Li, J., and Huang, W.: Formation of atmospheric molecular clusters of methanesulfonic acid–Diethylamine complex and its atmospheric significance, Atmospheric Environment, 226, 117404, https://doi.org/10.1016/j.atmosenv.2020.117404, 2020.

**Appendix**

**Table 1.** Cartesian coordinates of the IA dimer and trimer in the study of Rong et al., (2020) at the $\omega$B97X-D/6-311++G(3df,3pd) + aug-cc-pVDZ-PP with ECP28MDF (for I) level of theory.

$(IA)_2$

| Atoms | X | Y | Z |
| --- | --- | --- | --- |
| I | 1.605061 | -0.109352 | -0.267819 |
| O | 2.025006 | -1.477616 | 0.807848 |
| O | 0.441189 | 0.976342 | 0.730524 |
| O | 3.150010 | 1.046146 | 0.139201 |
| H | 3.315691 | 1.034733 | 1.088579 |
| I | -1.605061 | 0.109352 | 0.267819 |
| O | -0.441189 | -0.976343 | -0.730522 |
| O | -2.025004 | 1.477616 | -0.807848 |
| O | -3.150011 | -1.046145 | -0.139204 |
| H | -3.315691 | -1.034732 | -1.088582 |

$(IA)_3$

| Atoms | X | Y | Z |
| --- | --- | --- | --- |
| I | 1.545974 | -1.434287 | 0.250626 |
| O | 2.033030 | -0.476331 | -1.207134 |
| O | -0.211853 | -1.929524 | -0.055720 |
| O | 2.265224 | -3.140916 | -0.427111 |
| H | 2.210512 | -3.143192 | -1.389052 |
| I | 0.660386 | 1.874886 | -0.216130 |
| O | 0.871637 | 0.519496 | 1.066009 |
| O | -0.317464 | 3.088235 | 0.662250 |
| O | 2.415626 | 2.687111 | 0.162019 |
| H | 2.319948 | 3.214632 | 0.964029 |
| I | -2.180521 | -0.650525 | -0.103596 |
| O | -1.227504 | 0.715410 | -0.917840 |
| O | -2.135176 | -0.387443 | 1.665104 |
| O | -3.929096 | 0.185073 | -0.450347 |
| H | -4.015338 | 0.965703 | 0.109455 |

---

## Author Response (AR2)

Dear Editor,

Thank you very much for your handling our manuscript "**Molecular-level evidence for marine aerosol nucleation of iodic acid and methanesulfonic acid**" (MS No.: acp-2021-595) and giving us the opportunity to refine manuscript. According to reviewer's valuable comments, we have revised the manuscript carefully and listed the point-to-point responses to the reviewers' comments as below:

**Responses to Referee #3's comments:**

The authors have satisfactorily addressed my comments regarding the inconsistencies between their earlier study and the present study. I applaud the authors for deciding to prepare a corrigendum for Rong et al. 2020 paper and the corrections made to this paper concerning IA cluster structures.

Unfortunately, the author failed to address several key concerns which I had in the preface of my last review. After addressing the first-round reviews, the data quality of this manuscript is better. The revised data, to my opinion, clearly suggest that the IA-MSA particle formation mechanism quantitatively fails to explain both the field observations and laboratory investigations. This conflicts with the theme of this manuscript.

However, I think the data presented in this manuscript are still valuable if interpreted correctly. The authors should quantitatively compare their results with laboratory formation rates and conclude based on the comparison. They will find that their IA-MSA formation rates are too low (6 orders of magnitude lower) compared to the laboratory results and other species must be needed.

I understand that scholars tend to be reluctant to report negative results these days but I personally think a correctly interpreted negative result is certainly more valuable than misunderstood positive results. Thus, I urge the authors to make the suggested modifications.

**Response:** Thanks sincerely to the reviewer's helpful and professional comments. We have revised the manuscript accordingly. The detailed point-to-point responses are listed as follows.

**Major comments:**

**Comment 1.** In my previous review, I suggested that additional cluster geometries and free energies were needed to complete this study which was not adopted by the authors. The authors cited the statements from the ACDC manual and I attach the original whole paragraph below:

The criteria for clusters to grow out of the simulation, also known as boundary conditions, are essential parameters in an ACDC simulation. When a collision results in a cluster not included in the simulation set, it must be decided what to do with this product: is it reasonable to let it leave the simulation, or is it more likely that the cluster evaporates back to a smaller size? If the cluster composition can be assumed to be stable, i.e. molecular collisions are likely to dominate over evaporation, the cluster can be let out. As by default there is no information on the stabilities of clusters outside the system, the outgrowth criteria must be decided based on the existing data (molecular composition of the clusters along the main growth pathways, trends in the collision and evaporation rates inside the system...) and the best understanding (general chemistry of the clusters). An unreasonable choice of outgrowth criteria, on the other hand, may distort the simulation results.

The manual suggests that molecular collision should "dominate" evaporation. "Dominate" does not mean collision/evaporation equals one. So, lines starting line162 and SI are clearly misinterpretations. The authors can check e.g., Fig. 5.1 in the ACDC manual and relevant figures in e.g., Myllys et al. 2018, Elm et al. 2017 which all have much larger out-of-box collision to evaporation ratios.

The authors argue that their consideration of acid = 1e6 cm-3 is stringent and so the ratio equal to 0.2 is fine. This is not true. The authors should ensure that the out-of-box cluster has dominant collision over evaporation under all considered conditions (acid concentration, temperature etc.). A ratio of 10 is more adequate compared to 1. Thus, additional calculations are needed.

Additionally, I have a further question on the newly added equation SI. Can the authors specify where did they cite the equation from? The denominator is the sum of evaporation pathways which could be evaporating IA, MSA, IAMSA, IA2 etc but the numerator is the collision between IA and IA4MSA2? How about MSA and IA4MSA2? Please reconsider this.

**Response:** Thanks for the review's valuable suggestions. These suggestions are important for the rigor of the data presented.

**Item 1) from the reviewer:** The manual suggests that molecular collision should "dominate" evaporation. "Dominate" does not mean collision/evaporation equals one. So, lines starting line162 and SI are clearly misinterpretations. The authors can check e.g., Fig. 5.1 in the ACDC manual and relevant figures in e.g., Myllys et al. 2018, Elm et al. 2017 which all have much larger out-of-box collision to evaporation ratios.

**Response:** Indeed, the "dominate" in the ACDC manual does not correspond to a quantitative criterion. In our manuscript, the criterion of whether the collision/evaporation ratio exceeds 1 is based on the following statement in ACDC manual (section 5.1):

"If the relative evaporation rate decreases along the likely cluster growth pathway so that it is exceeded by the collision rate for the largest clusters, the system size can be considered sufficient."

Accordingly, the defined $\beta C / \sum \gamma > 1$ corresponds to the cluster evaporation rate being exceeded by the collision rate.

Furthermore, according to the reviewer's helpful suggestion, we have checked the mentioned Fig. 5.1 of the ACDC manual and relevant figures in e.g., Myllys et al. 2018, Elm et al. 2017. The larger out-of-box collision to evaporation ratios are presented in these studies. For example, in Fig. 5.1 of the ACDC manual, the value of $\beta_{(H2SO4)} C_{(H2SO4)} / \sum \gamma$ ($5H_2SO_4 \cdot 5NH_3$ cluster) is $2 \times 10^1$ in the $H_2SO_4$-$NH_3$ system at $T = 280$ K and $[H_2SO_4] = 5.0 \times 10^6$ molecules cm$^{-3}$ (a median value of the range of $[H_2SO_4]$, $10^6 \sim 10^7$ molecules cm$^{-3}$). Under similar conditions ($T = 278$ K and $[HIO_3] = 1.0 \times 10^7$ (a median value of $[HIO_3]$, $10^6 \sim 10^8$ molecules cm$^{-3}$ ), for the IA-MSA system, the $\beta_{(HIO3)} C_{(HIO3)} / \sum \gamma$ for $(IA)_4 \cdot (MSA)_2$ and $(IA)_6$ cluster is $2 \times 10^1$ and $7 \times 10^1$, respectively. The values of $\beta C / \sum \gamma$ presented are closely related to the employed temperature and nucleation monomer concentrations. Although our results are significantly lower than those for colliding with $NH_3$, namely $\beta_{(NH3)} C_{(NH3)} / \sum \gamma$ ($5H_2SO_4 \cdot 5NH_3$ cluster) in Fig. 5.1, colliding with acids ($\beta_{(H2SO4)} C_{(H2SO4)} / \sum \gamma$) would be a better boundary setting because of the instability of cluster with more base and less acid (Myllys et al., 2019).

To make the presented results more reliable, we performed additional cluster calculations and recalculated the relevant data according to the reviewer's professional suggestion, see the details in response for Item 2).

**Item 2) from the reviewer:** The authors argue that their consideration of acid = 1e6 cm-3 is stringent and so the ratio equal to 0.2 is fine. This is not true. The authors should ensure that the out-of-box cluster has dominant collision over evaporation under all considered conditions (acid concentration, temperature etc.). A ratio of 10 is more adequate compared to 1. Thus, additional calculations are needed.

**Response:** Thanks for the reviewer's careful review. As the reviewer expertly suggested, the ratio of cluster collision to evaporation is influenced by the considered conditions (temperature, acid concentration etc.). Generally, the lower temperatures and the higher acid concentrations employed, the larger ratio ($\beta C / \sum \gamma$) obtained.

The ratio of 0.2 presented in Figure S2 is the minimum value for $(IA)_6$ cluster under conditions of the highest studied temperature ($T = 298$ K) and lowest acid concentration ([IA] = $1.0 \times 10^6$ molecules cm$^{-3}$). But it is not fine to perform simulations under such conditions ($\beta_{IA} C_{IA} / \sum \gamma <1$). Hence, as shown in Figure 6 (d) of previous manuscript, we in fact simulated the case at 298 K with [IA] higher than $5.0 \times 10^6$ molecules cm$^{-3}$, at which the collision dominates over evaporation ($\beta_{IA} C_{IA} / \sum \gamma >1$). And at $T \leq 278$ K, the ACDC simulations were performed under condition of [IA] = $10^6 \sim 10^8$ molecules cm$^{-3}$, when $\beta_{IA} C_{IA} / \sum \gamma >1$. In all the ACDC simulations that have been performed, the out-of-box cluster has dominant collision over evaporation under the conditions employed.

Although, according to the ACDC manual, the system size is sufficient when the collision rate of the largest cluster exceeds its evaporation rate ($\beta C / \sum \gamma >1$), we agree with the reviewer that larger ratio than 1 is more adequate. Hence, according to the reviewer's helpful suggestion, we performed the additional cluster conformation calculations to find more stable $(IA)_6$ and $(IA)_4(MSA)_2$ clusters requested in the last Comment 3 from the reviewer. Here, we have doubled the conformational search range (5000 → 10,000), retaining 1000 structures from these initial 10,000 generations by ABCluster (Zhang and Dolg, 2015), on the basis of which a multi-step conformational search is further performed. The details of additional calculation as well as the resulting atomic coordinates for the new low-energy cluster are summarized in the appendix of this response.

Finally, inspired by the valuable suggestions of the reviewer, we find a $(IA)_6$ cluster with lower free energy. As show in Figure A1, the $\Delta G_{298K}$ of the new $(IA)_6$ cluster (-72.22 kcal mol$^{-1}$) is 2.10 kcal mol$^{-1}$ lower than the previous one (-70.12 kcal mol$^{-1}$).

[Figure]

$\Delta G_{298K} = -70.12$ kcal mol$^{-1}$        $\Delta G_{298K} = -72.22$ kcal mol$^{-1}$

Figure A1. The (IA)$_6$ cluster structure in the previous manuscript and the new (IA)$_6$ structure found after extended calculations. The black and green dashed lines indicate hydrogen and halogen bonds, respectively. The lengths of hydrogen and halogen bonds are given in Å.

Accordingly, we modified the structures of the (IA)$_6$ clusters in Figure S1 and updated the corresponding atomic coordinates in Table S7 in the revised Supporting Information (SI). And the newly-found (IA)$_6$ cluster is also mentioned in the method section. Further we recalculated the collision to evaporation ratio at 278 K and 298 K after changing the (IA)$_6$ clusters. The values of $\beta_{IA}C_{IA} / \sum\gamma$ increased from 0.2 to 5 at 298 K and from 7 to $3\times10^2$ at 278 K. The corrected data plots are presented in Figure 2 of the revised manuscript and Figure S2 and Figure S3 in SI. In addition, all data affected by changing (IA)$_6$ cluster were recalculated, and the corrected graphs and tables are highlighted in main text and SI.

**Item 3) from the reviewer:** Additionally, I have a further question on the newly added equation SI. Can the authors specify where did they cite the equation from? The denominator is the sum of evaporation pathways which could be evaporating IA, MSA, IAMSA, IA2 etc but the numerator is the collision between IA and IA4MSA2? How about MSA and IA4MSA2? Please reconsider this.

**Response:** We thank the reviewer's professional suggestions, which can make the data clearer for the readers. The newly added equation in SI was derived from the ratio of the collision rate to the evaporation rate, namely $\beta_{IA}C_{IA}\cdot C_{cluster} / \sum\gamma\cdot C_{cluster} = \beta_{IA}C_{IA} / \sum\gamma$. In fact, both cluster-monomer collisions and cluster-cluster collisions are considered in the performed ACDC simulations. Only collisions with IA monomers are presented here because the equilibrium concentration of clusters (e.g., (IA)$_1$(MSA)$_1$, (IA)$_2$) is significantly lower than that of IA monomers, and therefore collision of IA monomers is more possible than that of colliding clusters.

In addition, for the collision between MSA and (IA)$_4$(MSA)$_2$, it is considered and noted in

the previously revised manuscript (Line 167), Section S1 (SI), and the previous response to the Comment 3 (item 1). For example, in Line 167 of previously revised manuscript, the statement is: "Among these clusters, the largest $(IA)_4 \cdot (MSA)_2$ and $(IA)_6$ clusters ($\beta_{IA} C_{IA} / \sum \gamma > 1$) incline to collide with IA monomer (or MSA monomer) to grow out of the simulated system". Moreover, the boundary clusters in the performed simulations at 278 K contain $(IA)_7$, $(IA)_6 \cdot (MSA)_1$, $(IA)_4 \cdot (MSA)_3$, and $(IA)_5 \cdot (MSA)_2$ clusters, which were obtained by considering the collisions with IA or MSA monomer (Figure A2).

[Figure]

Figure A2. The baseline for setting the boundary clusters in ACDC simulations at 278K.

We thank the reviewer's valuable suggestion. Presenting the collision with MSA in a similar figure for IA is necessary for the clarity of the reader, as the statement alone is not clear. Hence, we have added the data of the colliding MSA monomers to Figure 2 of the revised manuscript, Figure S2 and Figure S3 in SI.
* * *
2. The authors need to include the formation rates from He et al. 2021 in their figures (e.g., Figure 3, 6). This way it will be apparent that there are 6 orders of magnitude differences between the CLOUD results and the present manuscript. While my suggestions concerning adding clusters will likely reduce the discrepancy, it will not resolve such a 6 orders of magnitude difference.

The author argued that both experiments and QC calculations have errors. However, such a huge error cannot be explained by systematic errors. QC+ACDC is a developed toolset and has been verified in multiple studies concerning e.g., SA + NH3 + DMA. Previous studies has reported reasonably well comparison with the CLOUD rates (e.g, Myllys et al. 2019; Myllys et al. 2016; Almeida et al. 2013; Olenius et al 2013). The 6 orders of magnitude discrepancy in

this study mush be due to other reasons.

The authors made a good point in their response that He et al. 2021 concluded that other iodine species (such as HIO2) are also critical in iodine particle formation processes. Therefore, the calculation of the pure-IA system in this manuscript is distinct from that of He et al. 2021. My revisit to the mentioned papers confirms their statement. Nevertheless, this affirms the fact that additional species are required to explain iodine + sulfur induced nucleation in marine and polar environments. I took a closer look at He et al 2021, particularly the part on how these HIO3 and HIO2 are formed. They seem to conclude that these species are simultaneously produced from iodine oxidation processes with relatively simple conditions analogous to the atmosphere. This would suggest that, besides HIO3, other iodine species are also present. Have the authors considered this possibility (e.g. is HIO2 important in the iodine nucleation?)?

**Response:** Thanks for the reviewer's valuable comments. These suggestions are helpful for refining the conclusions. The responses to each item are listed below.

**Item 1) from the reviewer:** The authors need to include the formation rates from He et al. 2021 in their figures (e.g., Figure 3, 6). This way it will be apparent that there are 6 orders of magnitude differences between the CLOUD results and the present manuscript. While my suggestions concerning adding clusters will likely reduce the discrepancy, it will not resolve such a 6 orders of magnitude difference.

**Response:** According to the reviewer's suggestion, the formation rates from He et al., 2021 has been included in Figure 3 of the revised manuscript. And we also discussed the differences between the simulated cluster formation rates and CLOUD results (Lines 191 – 197 of the revised manuscripts). For review convenience, we have copied the details as below:

"As shown in Fig. 3, the simulated $J$ of pure-IA nucleation (purple line) is much lower than the rate obtained from the CLOUD experiment (He et al. 2021). … Briefly, MSA can promote $J$ of IA clusters to a higher level, which is a reflection of the stabilizing effect of MSA on IA clusters. However, the $J$ of IA-MSA nucleation was still much less than the experimental results (He et al. 2021), even at a high [MSA] ($10^8$ molecules cm$^{-3}$). The large rate difference suggests that MSA stabilizes IA less efficiently than the potential iodine-containing components."

In addition, the additional clusters have been calculated and considered in the simulation of formation rate according to the reviewer's suggestion. As the reviewer expertly deduced, adding

clusters does not significantly reduce the discrepancy, which means that this is not the main reason and other iodine-containing acids potentially play a key role in nucleation.

**Item 2) from the reviewer:** The author argued that both experiments and QC calculations have errors. However, such a huge error cannot be explained by systematic errors. QC+ACDC is a developed toolset and has been verified in multiple studies concerning e.g., SA + NH3 + DMA. Previous studies has reported reasonably well comparison with the CLOUD rates (e.g, Myllys et al. 2019; Myllys et al. 2016; Almeida et al. 2013; Olenius et al 2013). The 6 orders of magnitude discrepancy in this study mush be due to other reasons.

**Response:** Indeed, in the mentioned studies, the QC+ACDC simulations shows a better fit with the CLOUD results. This is likely attributed to the fact that the employed nucleation precursors in the ACDC simulation are similar to those in the CLOUD chamber.

In contrast, our work aimed to study the stabilizing effect of MSA on IA, so iodine components other than IA were not considered. However, the iodine oxides and iodous acid are presented in the CLOUD experiment (He et al. 2021). We agree with the reviewer that the 5 ~ 6 orders of magnitude discrepancy should come from other causes, most likely the impact of iodine oxides and iodous acid.

**Item 3) from the reviewer:** The authors made a good point in their response that He et al. 2021 concluded that other iodine species (such as HIO2) are also critical in iodine particle formation processes. Therefore, the calculation of the pure-IA system in this manuscript is distinct from that of He et al. 2021. My revisit to the mentioned papers confirms their statement. Nevertheless, this affirms the fact that additional species are required to explain iodine + sulfur induced nucleation in marine and polar environments. I took a closer look at He et al 2021, particularly the part on how these HIO3 and HIO2 are formed. They seem to conclude that these species are simultaneously produced from iodine oxidation processes with relatively simple conditions analogous to the atmosphere. This would suggest that, besides HIO3, other iodine species are also present. Have the authors considered this possibility (e.g. is HIO2 important in the iodine nucleation?)?

**Response:** The reviewer's suggestions are far-sighted. Iodine species other than iodic acid (e.g., $I_2O_4$, $I_2O_5$, and $HIO_2$) were unfortunately not considered in the manuscript because the original purpose of this study was to explore the stabilizing effect of MSA on IA. Considering other iodine species would make it difficult to identify the role of MSA in IA nucleation, and would

result in a triple or even larger calculation efforts. Through this revised manuscript, especially inspired by the reviewers, we realized that other iodine species may play a more important role than MSA in IA cluster formation. Therefore, in order to simulate the real atmospheric environment and clarify the role of other iodine-containing components in IA nucleation, nucleation models including more iodine species, especially $HIO_2$, will be built in the future study.
* * *
3. In fact, the authors agree with my comments above by stating additional species, such as HIO2, iodine oxides, SA, NH3, amines can be important in marine and coastal particle formation processes too (lines 268 - 273). While I agree with the authors that marine environments are vast and different mechanisms can take place in different regions, the mechanism proposed in this study explains none of the mentioned field observations (Beck et al. 2021; Baccarini et al. 2020; Sipila et al. 2016).

The authors argue that their results do not conflict with Sipila et al. 2016 since they also suggest IA is important at Mace Head. However, at IA = 1E8, the authors calculated a formation rate of around 1e-2 cm-3 and even with additional MSA = 1E8, they get a formation rate of around 1 cm-3. However, the particle formation events at Mace Head are significantly stronger at the same acid concentration (Figure 1 in Sipila et al 2016 at IA = 1E8), sometimes reaching 1e7 cm-3 (O'dowd et al. 1999).

The same applies to Baccarini et al. 2020. Figure 1 in Baccarini et al. 2020 is at around 268 K which is not present in this manuscript. However, we can generously take the value from the results at 258K in this study (Figure 6c). With HIO3 at around 5e6, the formation rate is below 1e-5 cm-3 in this manuscript which indicates there should be no particle formation. This conflicts with Baccarini et al. 2020 since they observe strong particle formation events there.

The authors have removed the controversial Fig. 6 in the original manuscript which is good. In that figure, Beck et al 2021 also does not support the IA-MSA mechanism.

Considering these field observations and results from this manuscript, the correct derivation is that IA-MSA explains none of the mentioned field observations and additional species must be

needed. The authors are also encouraged to discuss the potential contributing species.

**Response:** This is a very helpful point – thanks for bringing it up. The correct presentation of negative results would be more valuable for future studies. We agree with the reviewer that this data discrepancy needs to be reduced by considering additional components. Hence, we discussed the potential species contributing to particle formation in section of Abstract, Results and Conclusion, respectively. The details are as follows:

a) **Abstract.**

   **Lines 14 - 16**: "… in the nucleation process. However, the nucleation rate of the IA-MSA mechanism is much lower than that of field observation, indicating that the role of additional nucleation precursors needs to be considered (e.g., $H_2SO_4$, $HIO_2$, $NH_3$ and amines)."

b) **Results.**

   **Lines 196 - 199:** "Briefly, MSA can promote $J$ of IA clusters to a higher level, which is a reflection of the stabilizing effect of MSA on IA clusters. However, the $J$ of IA-MSA nucleation was still much less than the experimental results (He et al. 2021), even at a high [MSA] ($10^8$ molecules cm$^{-3}$). The large rate difference suggests that MSA stabilizes IA less efficiently than the potential iodine-containing components."

   **Lines 274 – 280:** "Compared to the field observations at Mace Head (Sipila et al. 2016) and Arctic Sites (Beck et al. 2021), the rate of the IA-MSA mechanism is also significantly lower. This indicates that the contribution of MSA to IA particle formation under atmospheric conditions is relatively limited, and more efficient stabilizer for IA should be involved in the nucleating process, such as other iodine-containing components such as $HIO_2$ and iodine oxides ($I_2O_4$ and $I_2O_5$). Moreover, considering the complexity of the marine atmosphere, other non-iodine nucleation precursors (SA, $NH_3$, amines, etc.) may also affect the nucleation process, particularly with SA, because MSA and SA are both formed during the oxidation of DMS and coexist in the marine atmosphere. Therefore, in future studies, the influence of the above factors on the nucleation mechanism of marine aerosols will also be considered."

c) **Conclusion.**

   **Line 289 – 293:** "However, IA-MSA nucleation rates are far from sufficient to explain the field observations, indicating that additional essential precursors need to be considered (e.g.,

H₂SO₄, HIO₂, NH₃ and amines). Nucleation among these components is likely to be synergistic, with only varying magnitudes of contribution. For example, both SA and MSA originate from the oxidation of DMS, so their coexistence in the atmosphere may synergistically promote the formation of IA clusters, which is worthy of future studies."
* * *
4. L13: Before getting into the IA-MSA nucleation. 1) compare Pure IA results with experimental results and point out the difference. 2) explain the enhancement of Pure IA nucleation by MSA. 3) the combined effect is still too small compared to field observations. 4) additional species need to be considered (reasonably specify a few). The same applies to the conclusion part.

**Response:** Thanks for the reviewer's valuable suggestions, which are essential to refine the theme of this manuscript. Details of the revision for the Abstract and Conclusion sections are as follows.

a) **Abstract.**

**Line 12 - 16:** "The findings show that the pure-IA nucleation rate was much lower than the results of CLOUD experiments. MSA can promote IA cluster formation through stabilizing IA via both hydrogen and halogen bonds, especially under conditions with lower temperatures, sparse IA and rich MSA. However, the nucleation rate of the IA-MSA mechanism is much lower than that of field observation, indicating that the effect of additional nucleation precursors needs to be considered (e.g., H₂SO₄, HIO₂, NH₃ and amines)."

b) **Conclusion**

**Line 284 - 293:** "…Atmospheric Cluster Dynamics Code (ACDC). The results suggest that the self-nucleation rate of IA is much lower than that of CLOUD experiment, indicating stabilizers are essential for IA nucleation process. We find that MSA can stabilize IA cluster via both hydrogen and halogen bonds, and thus promote IA cluster formation rate, especially in low-temperature environments with sparse IA and rich MSA. The corresponding IA-MSA nucleating mechanism can be described by two distinct pathways: i) pure-IA cluster formation and ii) IA-MSA cluster formation, indicating that IA and MSA can jointly nucleate. The IA-MSA nucleation is highly dependent on the distribution of MSA and IA in the marine atmosphere. However, IA-MSA nucleation rates are far from

sufficient to explain the field observations, indicating that additional essential precursors need to be considered (e.g., $H_2SO_4$, $HIO_2$, $NH_3$ and amines). Nucleation among these components is likely to be synergistic, with only varying magnitudes of contribution. For example, both SA and MSA originate from the oxidation of DMS, so their coexistence in the atmosphere may synergistically promote the formation of IA clusters, which is worthy of future studies."
* * *
**Minor comments:**

L40: My read does not find that He et al. 2021 suggests the involvement of I2O5 and I2O4 in the nucleation? It looks to me that they cannot conclude the role of I2O4/I2O5. But maybe the authors can refer me to the relevant sentences.

**Response:** The statement in the work of He et al. 2021 regarding the involvement of $I_2O_5$ and $I_2O_4$ in nucleation is in the section of "Particle Composition". Additionally, the role of $I_2O_4$ and $I_2O_5$ in nucleation is presented in the Figure 3 (He et al. 2021). For the convenience of the reviewer, the relevant sentences were copied as below.

"The measurements presented in Fig. 1B provide strong evidence that $HIO_3$ drives the growth of iodic particles above 1.8 nm. However, we have seen that additional iodine compounds play important roles during nucleation: $HIO_2$ for neutral nucleation, and the formation of iodine oxides—$I_2O_5$ and $I_2O_4$ —in the charged and neutral clusters, respectively (Fig. 2)."
* * *
L67: Aside from the difference in calculation methods, the authors should also write clearly that the geometries of IA2 and IA3 are different from what they have presented in their response.

**Response:** Thanks for the reviewer's helpful suggestion. The differences in the IA2 and IA3 cluster have been corrected. The differences of employed cluster structures are also described in the Methods section of the revised manuscript.
* * *
In addition, to further weaken the impact of IA-MSA mechanism, the additional changes were made as following:

Thanks to the reviewer's very important suggestion, we accordingly refine the theme of this manuscript and change the title as following.

**Original Title**: Molecular-level evidence for marine aerosol nucleation of iodic acid and

methanesulfonic acid.

**Corrected:** Molecular-level nucleation mechanism of iodic acid and methanesulfonic acid.

**Lines 16 – 17**: "The IA-MSA nucleation mechanism revealed in this study may help to elucidate some missing sources of marine NPF."
**Corrected:** "The IA-MSA nucleation mechanism revealed in this study may help to gain insight into the joint effect of marine sulfur- and iodine-containing components on marine NPF."

**Line 196:** Briefly, MSA can promote $J$ of IA clusters to a higher level, which may help explain the rapid formation of IA-involved particles in some marine NPF.
**Corrected:** "Briefly, MSA can promote $J$ of IA clusters to a higher level, which is a reflection of the stabilizing effect of MSA on IA clusters."

Thanks again for the reviewer's professional and carefulness review. Accordingly, we have tried our best to improve the manuscript.

Sincerely Yours,
Prof. Xiuhui Zhang

**Appendix.**

**Additional calculation details.**

First, the artificial bee algorithm combining the UFF force field (Rappe et al., 1992) was adopted to yield 1000 initial-guess structures from 10,000 generations by ABCluster software (Zhang and Dolg, 2015). After pre-optimization by the PM7 method (Stewart, 2013) with MOPAC2016 (James J. P. Stewart, 2016), the 100 lower-energy structures were left for further optimization at $\omega$B97X-D/6-31+G* + Lanl2DZ (for iodine) level of theory. The final cluster structure were optimized at $\omega$B97X-D/6-311++G(3df,3pd) (H, C, O and S) (Francl et al., 1982) + aug-cc-pVTZ-PP with ECP28MDF (for I) (Peterson et al., 2003).

All the additional calculation result files for each step are available at the following URL: https://www.icloud.com/iclouddrive/011Z9EW6_j0-XVvsJ0AzSWvZQ#IA-MSA_.Additional_Calculation)

**Table A1.** Atomic coordinates of the $(IA)_6$ cluster in the previous manuscript and in this revised manuscript.

**(IA)$_6$ (Previous):**

| Atoms | X | Y | Z |
|-------|-----------|------------|------------|
| I | 2.0142330 | -1.9502430 | 0.1062340 |
| O | 3.5813460 | -2.3077960 | -0.6609640 |
| O | 1.2147740 | -0.8403890 | -1.0583760 |
| O | 1.0568390 | -3.5830250 | -0.4682790 |
| H | 1.3933920 | -3.8971680 | -1.3144030 |
| I | 4.0616510 | 1.1772010 | 0.5078450 |
| O | 3.6480600 | 1.4144440 | -1.2121400 |
| O | 2.9102220 | -0.0865740 | 1.1649480 |
| O | 5.4562590 | -0.1115870 | 0.2184390 |
| H | 5.0709030 | -0.9080240 | -0.1938550 |
| I | -1.9509040 | -2.0075870 | 1.2785530 |
| O | -0.3728890 | -1.2094910 | 1.5384160 |
| O | -2.2436270 | -1.9045460 | -0.4983290 |
| O | -1.2923970 | -3.8204880 | 1.2425070 |
| H | -0.5887520 | -3.9170540 | 0.5764410 |
| I | -3.5541860 | 1.5105690 | 0.6518740 |
| O | -4.9348430 | 2.0979990 | 1.5719390 |
| O | -3.0518940 | 0.0320430 | 1.5940770 |
| O | -4.5205320 | 0.6939320 | -0.7634460 |
| H | -3.9463030 | 0.3947540 | -1.5156220 |

| | | | |
|---|---|---|---|
| I | 0.6743480 | 1.8105790 | 0.3807700 |
| O | 2.2811220 | 2.5127150 | 0.9739480 |
| O | -0.4695840 | 3.0879240 | 0.7922740 |
| O | 1.0853560 | 2.2419120 | -1.4546750 |
| H | 2.0513300 | 2.0382580 | -1.5418810 |
| I | -1.2858540 | -0.3923390 | -2.1824110 |
| O | -1.2960600 | 0.8181160 | -0.8255000 |
| O | -2.9363330 | -0.1593290 | -2.8052690 |
| O | -0.3635120 | 0.7547110 | -3.4115000 |
| H | 0.1587040 | 1.3911450 | -2.8870510 |

**(IA)₆ (New):**

| Atoms | X | Y | Z |
|---|---|---|---|
| I | 2.2808360 | -0.2753710 | -2.1881650 |
| O | 2.9719550 | -0.8720370 | -0.6598240 |
| O | 0.7608060 | 0.5949180 | -1.7316370 |
| O | 3.3326370 | 1.3014170 | -2.4413400 |
| H | 2.9510610 | 2.0411620 | -1.9103970 |
| I | 0.6413140 | 2.4184810 | -0.0083520 |
| O | -0.8608180 | 3.0004440 | -0.8644300 |
| O | 1.9683430 | 3.1662190 | -0.9376500 |
| O | 0.6243090 | 3.7769460 | 1.3626620 |
| H | 1.1942090 | 4.4944990 | 1.0581850 |
| I | -2.7890380 | 1.8068790 | -0.2953190 |
| O | -1.7709380 | 1.3170510 | 1.1505700 |
| O | -2.4200660 | 0.6546250 | -1.5917610 |
| O | -4.3947320 | 0.9695600 | 0.3615050 |
| H | -4.1653500 | 0.0378670 | 0.5465660 |
| I | -1.4589660 | -1.0321300 | 1.8638060 |
| O | -0.1031870 | -0.9421340 | 0.6382260 |
| O | -2.8761320 | -1.3267160 | 0.8107060 |
| O | -1.1795240 | -2.8714060 | 2.2946450 |
| H | -1.2063490 | -3.3478930 | 1.4394730 |
| I | -0.9196570 | -2.0764910 | -1.5666630 |
| O | -1.0281490 | -3.5252090 | -0.5362830 |
| O | 0.7784970 | -2.1746890 | -2.1884340 |
| O | -1.7723570 | -2.8490540 | -3.1160810 |
| H | -1.6671070 | -3.8081140 | -3.0746010 |
| I | 2.1173610 | -0.7524830 | 2.0447950 |
| O | 0.8024440 | -0.7625620 | 3.2681640 |
| O | 2.3028120 | 0.9848650 | 1.6935780 |
| O | 3.6177060 | -0.9370970 | 3.2444890 |
| H | 3.6566860 | -0.1695010 | 3.8285950 |

**References:**

Myllys, N., Elm, J., Halonen, R., Kurtén, T., and Vehkamäki, H.: Coupled Cluster Evaluation of the Stability of Atmospheric Acid–Base Clusters with up to 10 Molecules, J. Phys. Chem. A, 120, 621–630, https://doi.org/10.1021/acs.jpca.5b09762, 2016.

Myllys, N., Ponkkonen, T., Passananti, M., Elm, J., Vehkamäki, H., and Olenius, T.: Guanidine: A Highly Efficient Stabilizer in Atmospheric New-Particle Formation, 122, 4717–4729, https://doi.org/10.1021/acs.jpca.8b02507, 2018.

Myllys, N., Chee, S., Olenius, T., Lawler, M., and Smith, J.: Molecular-Level Understanding of Synergistic Effects in Sulfuric Acid–Amine–Ammonia Mixed Clusters, J. Phys. Chem. A, 123, 2420–2425, https://doi.org/10.1021/acs.jpca.9b00909, 2019.

Rong, H., Liu, J., Zhang, Y., Du, L., Zhang, X., and Li, Z.: Nucleation mechanisms of iodic acid in clean and polluted coastal regions, Chemosphere, 253, 126743, https://doi.org/10.1016/j.chemosphere.2020.126743, 2020.

Beck, L. J., Sarnela, N., Junninen, H., Hoppe, C. J. M., Garmash, O., Bianchi, F., Riva, M., Rose, C., Peräkylä, O., Wimmer, D., Kausiala, O., Jokinen, T., Ahonen, L., Mikkilä, J., Hakala, J., He, X., Kontkanen, J., Wolf, K. K. E., Cappelletti, D., Mazzola, M., Traversi, R., Petroselli, C., Viola, A. P., Vitale, V., Lange, R., Massling, A., Nøjgaard, J. K., Krejci, R., Karlsson, L., Zieger, P., Jang, S., Lee, K., Vakkari, V., Lampilahti, J., Thakur, R. C., Leino, K., Kangasluoma, J., Duplissy, E., Siivola, E., Marbouti, M., Tham, Y. J., Saiz-Lopez, A., Petäjä, T., Ehn, M., Worsnop, D. R., Skov, H., Kulmala, M., Kerminen, V., and Sipilä, M.: Differing Mechanisms of New Particle Formation at Two Arctic Sites, Geophys Res Lett, 48, https://doi.org/10.1029/2020GL091334, 2021.

Sipilä, M., Sarnela, N., Jokinen, T., Henschel, H., Junninen, H., Kontkanen, J., Richters, S., Kangasluoma, J., Franchin, A., Peräkylä, O., Rissanen, M. P., Ehn, M., Vehkamäki, H., Kurten, T., Berndt, T., Petäjä, T., Worsnop, D., Ceburnis, D., Kerminen, V.-M., Kulmala, M., and O'Dowd, C.: Molecular-scale evidence of aerosol particle formation via sequential addition of HIO3, Nature, 537, 532–534, https://doi.org/10.1038/nature19314, 2016.

Baccarini, A., Karlsson, L., Dommen, J., Duplessis, P., Vüllers, J., Brooks, I. M., Saiz-Lopez,

A., Salter, M., Tjernström, M., Baltensperger, U., Zieger, P., and Schmale, J.: Frequent new particle formation over the high Arctic pack ice by enhanced iodine emissions, Nature Communications, 11, 4924, https://doi.org/10.1038/s41467-020-18551-0, 2020.

He, X.-C., Tham, Y. J., Dada, L., Wang, M., Finkenzeller, H., Stolzenburg, D., Iyer, S., Simon, M., Kürten, A., Shen, J., Rörup, B., Rissanen, M., Schobesberger, S., Baalbaki, R., Wang, D. S., Koenig, T. K., Jokinen, T., Sarnela, N., Beck, L. J., Almeida, J., Amanatidis, S., Amorim, A., Ataei, F., Baccarini, A., Bertozzi, B., Bianchi, F., Brilke, S., Caudillo, L., Chen, D., Chiu, R., Chu, B., Dias, A., Ding, A., Dommen, J., Duplissy, J., El Haddad, I., Gonzalez Carracedo, L., Granzin, M., Hansel, A., Heinritzi, M., Hofbauer, V., Junninen, H., Kangasluoma, J., Kemppainen, D., Kim, C., Kong, W., Krechmer, J. E., Kvashin, A., Laitinen, T., Lamkaddam, H., Lee, C. P., Lehtipalo, K., Leiminger, M., Li, Z., Makhmutov, V., Manninen, H. E., Marie, G., Marten, R., Mathot, S., Mauldin, R. L., Mentler, B., Möhler, O., Müller, T., Nie, W., Onnela, A., Petäjä, T., Pfeifer, J., Philippov, M., Ranjithkumar, A., Saiz-Lopez, A., Salma, I., Scholz, W., Schuchmann, S., Schulze, B., Steiner, G., Stozhkov, Y., Tauber, C., Tomé, A., Thakur, R. C., Väisänen, O., Vazquez-Pufleau, M., Wagner, A. C., Wang, Y., Weber, S. K., Winkler, P. M., Wu, Y., Xiao, M., Yan, C., Ye, Q., Ylisirniö, A., Zauner-Wieczorek, M., Zha, Q., Zhou, P., Flagan, R. C., Curtius, J., Baltensperger, U., Kulmala, M., Kerminen, V.-M., Kurtén, T., et al.: Role of iodine oxoacids in atmospheric aerosol nucleation, 371, 589–595, https://doi.org/10.1126/science.abe0298, 2021.

Elm, J., Passananti, M., Kurtén, T., and Vehkamäki, H.: Diamines Can Initiate New Particle Formation in the Atmosphere, J. Phys. Chem. A, 121, 6155–6164, https://doi.org/10.1021/acs.jpca.7b05658, 2017.

McGrath, M. J., Olenius, T., Ortega, I. K., Loukonen, V., Paasonen, P., Kurtén, T., Kulmala, M., and Vehkamäki, H.: Atmospheric Cluster Dynamics Code: a flexible method for solution of the birth-death equations, Atmos. Chem. Phys., 12, 2345–2355, https://doi.org/10.5194/acp-12-2345-2012, 2012.

Olenius, T., Kupiainen-Määttä, O., Ortega, I. K., Kurtén, T., and Vehkamäki, H.: Free energy barrier in the growth of sulfuric acid–ammonia and sulfuric acid–dimethylamine clusters, 13, 2013.

Almeida, J., Schobesberger, S., Kürten, A., Ortega, I. K., Kupiainen-Määttä, O., Praplan, A. P.,

Adamov, A., Amorim, A., Bianchi, F., Breitenlechner, M., David, A., Dommen, J., Donahue, N. M., Downard, A., Dunne, E., Duplissy, J., Ehrhart, S., Flagan, R. C., Franchin, A., Guida, R., Hakala, J., Hansel, A., Heinritzi, M., Henschel, H., Jokinen, T., Junninen, H., Kajos, M., Kangasluoma, J., Keskinen, H., Kupc, A., Kurtén, T., Kvashin, A. N., Laaksonen, A., Lehtipalo, K., Leiminger, M., Leppä, J., Loukonen, V., Makhmutov, V., Mathot, S., McGrath, M. J., Nieminen, T., Olenius, T., Onnela, A., Petäjä, T., Riccobono, F., Riipinen, I., Rissanen, M., Rondo, L., Ruuskanen, T., Santos, F. D., Sarnela, N., Schallhart, S., Schnitzhofer, R., Seinfeld, J. H., Simon, M., Sipilä, M., Stozhkov, Y., Stratmann, F., Tomé, A., Tröstl, J., Tsagkogeorgas, G., Vaattovaara, P., Viisanen, Y., Virtanen, A., Vrtala, A., Wagner, P. E., Weingartner, E., Wex, H., Williamson, C., Wimmer, D., Ye, P., Yli-Juuti, T., Carslaw, K. S., Kulmala, M., Curtius, J., Baltensperger, U., Worsnop, D. R., Vehkamäki, H., and Kirkby, J.: Molecular understanding of sulphuric acid–amine particle nucleation in the atmosphere, Nature, 502, 359–363, https://doi.org/10.1038/nature12663, 2013.

O'Dowd, C., McFiggans, G., Creasey, D. J., Pirjola, L., Hoell, C., Smith, M. H., Allan, B. J., Plane, J. M. C., Heard, D. E., Lee, J. D., Pilling, M. J., and Kulmala, M.: On the photochemical production of new particles in the coastal boundary layer, Geophysical Research Letters, 26, 1707–1710, https://doi.org/10.1029/1999GL900335, 1999.

Francl, M. M., Pietro, W. J., Hehre, W. J., Binkley, J. S., Gordon, M. S., DeFrees, D. J., and Pople, J. A.: Self-consistent molecular orbital methods. XXIII. A polarization-type basis set for second-row elements, J. Chem. Phys., 77, 3654–3665, https://doi.org/10.1063/1.444267, 1982.

James J. P. Stewart: MOPAC2016, Colo. SpringsCOUSA, 2016.

Myllys, N., Kubečka, J., Besel, V., Alfaouri, D., Olenius, T., Smith, J. N., and Passananti, M.: Role of base strength, cluster structure and charge in sulfuric-acid-driven particle formation, Atmospheric Chem. Phys., 19, 9753–9768, https://doi.org/10.5194/acp-19-9753-2019, 2019.

Peterson, K. A., Figgen, D., Goll, E., Stoll, H., and Dolg, M.: Systematically convergent basis sets with relativistic pseudopotentials. II. Small-core pseudopotentials and correlation consistent basis sets for the post- $d$ group 16–18 elements, J. Chem. Phys., 119, 11113–11123, https://doi.org/10.1063/1.1622924, 2003.

Rappe, A. K., Casewit, C. J., Colwell, K. S., Goddard, W. A., and Skiff, W. M.: UFF, a full periodic table force field for molecular mechanics and molecular dynamics simulations, J. Am. Chem. Soc., 114, 10024–10035, https://doi.org/10.1021/ja00051a040, 1992.

Stewart, J. J. P.: Optimization of parameters for semiempirical methods VI: more modifications to the NDDO approximations and re-optimization of parameters, J. Mol. Model., 19, 1–32, https://doi.org/10.1007/s00894-012-1667-x, 2013.

Zhang, J. and Dolg, M.: ABCluster: the artificial bee colony algorithm for cluster global optimization, Phys. Chem. Chem. Phys., 17, 24173–24181, https://doi.org/10.1039/C5CP04060D, 2015.